# The origin of ultrahigh piezoelectricity in relaxor-ferroelectric solid solution crystals

Fei Li[1,2], Shujun Zhang[1,3], Tiannan Yang[1], Zhuo Xu[2], Nan Zhang[2], Gang Liu[4,5], Jianjun Wang[1], Jianli Wang[3], Zhenxiang Cheng[3], Zuo-Guang Ye[6,2], Jun Luo[7], Thomas R. Shrout[1] & Long-Qing Chen[1]

The discovery of ultrahigh piezoelectricity in relaxor-ferroelectric solid solution single crystals is a breakthrough in ferroelectric materials. A key signature of relaxor-ferroelectric solid solutions is the existence of polar nanoregions, a nanoscale inhomogeneity, that coexist with normal ferroelectric domains. Despite two decades of extensive studies, the contribution of polar nanoregions to the underlying piezoelectric properties of relaxor ferroelectrics has yet to be established. Here we quantitatively characterize the contribution of polar nanoregions to the dielectric/piezoelectric responses of relaxor-ferroelectric crystals using a combination of cryogenic experiments and phase-field simulations. The contribution of polar nanoregions to the room-temperature dielectric and piezoelectric properties is in the range of 50–80%. A mesoscale mechanism is proposed to reveal the origin of the high piezoelectricity in relaxor ferroelectrics, where the polar nanoregions aligned in a ferroelectric matrix can facilitate polarization rotation. This mechanism emphasizes the critical role of local structure on the macroscopic properties of ferroelectric materials.

[1] Department of Materials Science and Engineering, Materials Research Institute, Pennsylvania State University, University Park, Pennsylvania 16802, USA. [2] Electronic Materials Research Laboratory, Key Laboratory of the Ministry of Education and International Center for Dielectric Research, Xi'an Jiaotong University, Xi'an 710049, China. [3] Institute for Superconducting and Electronic Materials, Australian Institute of Innovative Materials, University of Wollongong, Wollongong, New South Wales 2500, Australia. [4] High Pressure Synergetic Consortium, Geophysical Laboratory, Carnegie Institute of Washington, Argonne, Illinois 60439, USA. [5] Center for High Pressure Science and Technology Advanced Research, Shanghai 201203, China. [6] Department of Chemistry and 4D LABS, Simon Fraser University, Burnaby, British Columbia, Canada V5A 1S6. [7] TRS Technologies Inc., 2820 E. College Avenue, State College, Pennsylvania 16801, USA. Correspondence and requests for materials should be addressed to S.Z. (email: shujun@uow.edu.au) or to L.-Q.C. (email: lqc3@psu.edu).

Perovskite ferroelectrics (general formula, $ABO_3$) exhibit the highest electromechanical activity among all known piezoelectrics. The enhanced piezoelectric response of perovskite ferroelectrics are generally associated with the long-range cooperative phenomena near morphotropic phase boundaries (MPBs)[1–5]. In proximity of an MPB, different ferroelectric phases possess similar energies, leading to facilitated variation of polarization and strain under an external stimulus. One of the most remarkable breakthroughs in perovskite ferroelectrics was the discovery of ultrahigh piezoelectricity ($d_{33}^* = 1,500–2,500\,\mathrm{pC\,N^{-1}}$) and electromechanical coupling factors ($k_{33}^* > 0.9$) in domain-engineered relaxor–$PbTiO_3$ (PT) solid solution crystals with MPB compositions, for example, $Pb(Mg_{1/3}Nb_{2/3})O_3$–PT (PMN–PT) and $Pb(Zn_{1/3}Nb_{2/3})O_3$–PT (PZN–PT) crystals[6–12].

By considering the diversity and instability of ferroelectric phases at MPBs, a number of qualitative mechanisms have been proposed to elucidate the high piezoelectric activity in relaxor–PT crystals. The corresponding mechanisms include 'electric field-induced phase transition'[6], 'ease of polarization rotation via a monoclinic phase'[13,14], 'giant electromechanical response as a critical phenomenon'[15], 'adaptive domain structure'[16] and so on. However, these mechanisms fail to explain why relaxor-ferroelectric solid solutions exhibit significantly higher piezoelectricity when compared with non-relaxor-based MPB ferroelectrics, for example, $Pb(Zr_xTi_{1-x})O_3$ crystals (500–1,000 $\mathrm{pC\,N^{-1}}$ for $x$ around 0.5).

To further clarify the origin of ultrahigh piezoelectricity in relaxor–PT crystals, the microstructural characteristics of relaxor ferroelectrics should be surveyed first. Relaxors, for example, PMN, are characterized by cation disorder on the nanoscale[17–20],

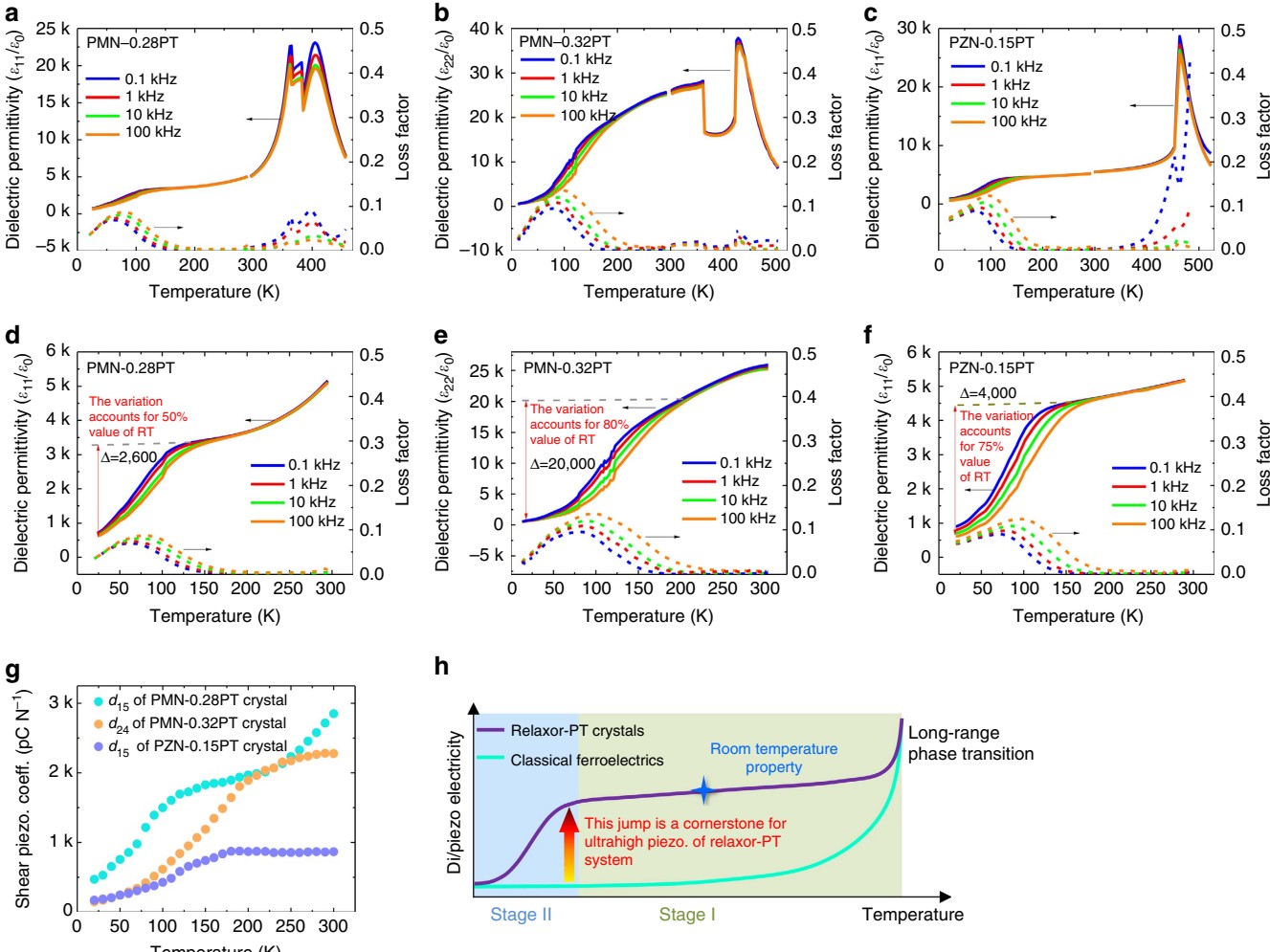

**Figure 1 | The temperature dependences of transverse dielectric ($\varepsilon_{11}$ and $\varepsilon_{22}$) and shear piezoelectric ($d_{15}$ and $d_{24}$) properties for single-domain relaxor–PT crystals.** (**a–c**) Dielectric permittivities for the rhombohedral PMN-0.28PT, orthorhombic PMN-0.32PT and tetragonal PZN-0.15PT crystals, respectively. In the PMN-0.28PT crystals, a rhombohedral–tetragonal and a tetragonal–cubic phase transitions exist at 368 and 405 K, respectively. In the PMN-0.32PT crystal, an orthorhombic–tetragonal and a tetragonal–cubic phase transitions occur at 360 and 430 K, respectively. In the PZN-0.15PT crystal, a tetragonal–cubic phase transition exists at 468 K. (**d–f**) Enlarged low-temperature sections (20–300 K) for **a–c**, respectively. (**g**) The measured shear piezoelectric coefficients for the three relaxor–PT crystals as a function of temperature. (**h**) A schematic plot showing the major finding of this work for the enhancement of dielectric/piezoelectric properties in relaxor–PT crystals (the temperature-dependent piezoelectric/dielectric responses of the classical ferroelectrics is inferred from phenomenological theory and first-principle calculations of $Pb(Zr_{0.5}Ti_{0.5})O_3$ (refs 50,51)). For comparison, the corresponding temperature dependences of the longitudinal dielectric permittivity $\varepsilon_{33}$ of single-domain PMN-0.28PT, PMN-0.32PT and PZN-0.15PT crystals are shown in Supplementary Fig. 1. For orthorhombic crystals, $\varepsilon_{11}$ and $\varepsilon_{22}$ are two independent tensors. The permittivity $\varepsilon_{11}$ versus temperature of the orthorhombic PMN-0.32PT crystal is given in Supplementary Fig. 2. For the PZN-0.15PT crystal, the temperature dependence of the reciprocal of relative dielectric constant is given in Supplementary Fig. 3, for identifying its relaxor characteristic.

leading to random fields[19,20] and local phase fluctuations[17,18]. These factors lead to a unique characteristic of relaxor-based ferroelectrics in contrast to classical ferroelectrics, that is, the presence of polar nanoregions (PNRs)[17–21], which are believed to be responsible for the high dielectric properties of relaxors[17,18,22]. Therefore, there have been numerous studies to understand the relationship between the high piezoelectric properties and PNRs in relaxor–PT systems[23–27]. For example, Xu et al.[23] proposed that the softening of the transverse acoustic mode was due to the existence of PNRs, while Manley et al.[24] further demonstrated that aligning PNR vibrational modes by a poling electric field can enhance the phonon softening. However, there have been no sufficient evidence from piezoelectric/dielectric measurements to substantiate the contribution of PNRs, which highly impedes the understanding of the corresponding mechanism(s) for ultrahigh piezoelectricity in relaxor–PT systems.

In this work, we obtain crucial experimental evidences for elucidating the contribution of PNRs to the piezoelectric activity in relaxor–PT crystals through cryogenic measurements, whereupon the PNR's contribution accounts for 50–80% of the room-temperature dielectric and piezoelectric properties. In addition, we perform phase-field simulations of dielectric/piezoelectric responses of ferroelectric domains in the presence of PNRs and propose a mesoscale mechanism that successfully explains the ultrahigh dielectric/piezoelectric properties and the corresponding temperature dependence for relaxor–PT crystals.

## Results

**Crystal samples.** Our investigations focused on the transverse dielectric and shear piezoelectric responses of single-domain relaxor–PT crystals. As listed in Supplementary Table 1, the transverse dielectric and shear piezoelectric properties are significantly larger than their longitudinal counterparts, and thus play a dominant role in the high electromechanical performance of relaxor–PT crystals. It should be noted that the large

longitudinal piezoelectric properties in domain-engineered relaxor–PT crystals originate from the high shear piezoelectric response in the corresponding single-domain state[28–30]. To demonstrate the generality of our results and the proposed mechanism, we investigated three different types of phases with all of them single-domain crystals: (1) rhombohedral [111]-poled PMN–0.28PT; (2) orthorhombic [011]-poled PMN–0.32PT; and (3) tetragonal [001]-poled PZN–0.15PT crystals.

**Cryogenic experiments.** Figure 1 shows the transverse dielectric permittivity ($\varepsilon_{11}/\varepsilon_0$ or $\varepsilon_{22}/\varepsilon_0$) versus temperature for single-domain rhombohedral PMN–0.28PT, orthorhombic PMN–0.32PT and tetragonal PZN–0.15PT crystals, respectively, measured over a frequency range of ($10^2 \sim 10^5$ Hz). Above room temperature, several phase transitions are found to exist. For example, rhombohedral-to-tetragonal ($T_{rt}$: 368 K), orthorhombic-to-tetragonal ($T_{ot}$: 360 K) and tetragonal-to-cubic ($T_{tc}$: 468 K) phase transitions are observed in the PMN–0.28PT, PMN–0.32PT and PZN–0.15PT crystals, respectively. These phase transitions destroy the single-domain states of the crystals, and thus our studies are focused on temperatures well below these phase transitions.

The temperature-dependent dielectric behaviour of the single-domain phases can be divided into two stages: stage I ($T > 200$ K) and stage II ($T < 200$ K). In stage I, the frequency dispersion of the dielectric permittivity is minimal, and the variation of dielectric response versus temperature can be explained by the Landau–Devonshire phenomenological theory, as previously reported[30]. In stage II, a drastic increase in transverse dielectric permittivity, $\varepsilon_{11}/\varepsilon_0$ and/or $\varepsilon_{22}/\varepsilon_0$, is observed in all three crystals with increasing temperature, accompanied by dielectric relaxation, that is, frequency dispersion of permittivity and loss maxima. As expected, the shear piezoelectric coefficients $d_{15}/d_{24}$, which can be expressed as $d_{15/24} = 2Q_{55/44}\varepsilon_{11}P_S$, follow similar temperature-dependent behaviour to the dielectric permittivities in stage II (Fig. 1g), since variations in spontaneous polarization $P_S$

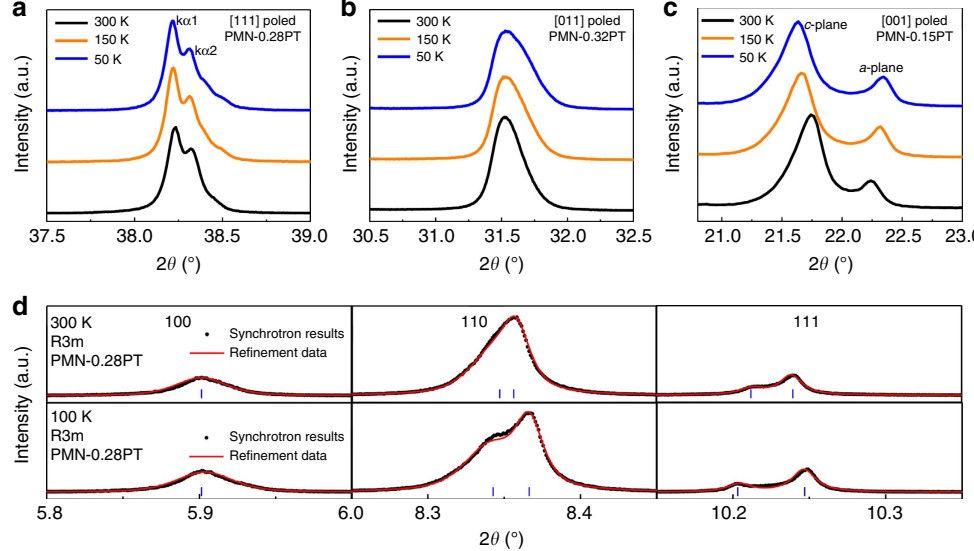

**Figure 2 | X-ray diffraction experiments of relaxor–PT at cryogenic temperatures.** (**a–c**) X-ray diffraction patterns for single-domain crystals: the (111) surface of [111]-poled PMN–0.28PT; the (011) surface of [011]-poled PMN–0.32PT; and the (001) surface of [001]-poled PZN–0.15PT crystals, respectively. For all the three crystals, the X-ray diffraction peaks do not show any anomaly over the temperature range of 50–300 K, indicating the lack of any long-range ferroelectric phase transition at cryogenic temperatures. Note: the peak splitting in PZN–0.15PT is due to an incomplete single-domain state, especially around the surface of the sample[49]. Because of the high lattice parameter ratio $c/a$ (1.023) and the associated mechanical clamping, a fully single-domain state is very difficult to obtain. (**d**) A selected synchrotron X-ray diffraction pattern for PMN–0.28PT powders (ground from crystal samples), where the Rietveld-refinement parameters are given in Supplementary Table 2. The synchrotron X-ray diffraction results indicate that the long-range structure of PMN–0.28PT is $R3m$ from 300 down to 100 K.

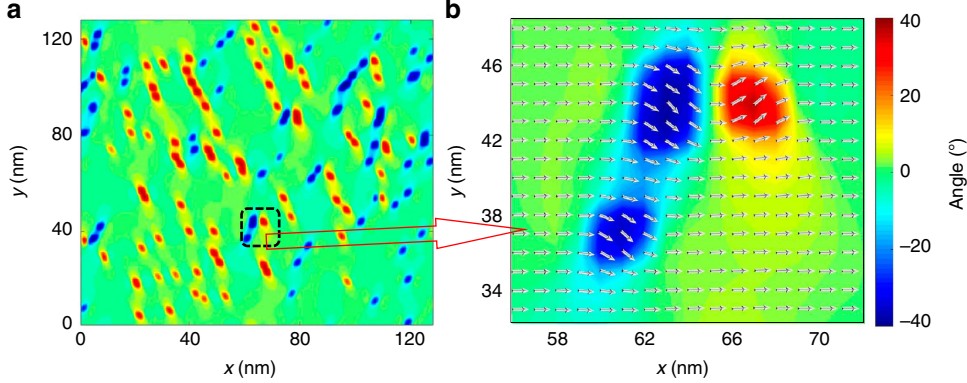

**Figure 3 | Phase-field-simulated microstructures for a [100]-poled PNR-ferroelectric composite at 100 K. (b)** The enlarged square area in **a**, where the polar vectors of every grid are shown. The x and y axes represent the [100] and [010] crystallographic directions, respectively. The unit for x and y axes is nanometre. The colour bar denotes the angle between the polar vector of the grids and the [100] direction.

(Supplementary Fig. 4) and electrostrictive coefficients $Q_{55/44}$ are minimal in this temperature range[31]. It should be noted that the drastic increase in transverse dielectric permittivities observed in stage II account for 50–80% of their room-temperature values, as depicted in Fig. 1d–f, hence they play a dominant role in the ultrahigh room-temperature properties of relaxor–PT crystals.

The enhancement of dielectric/piezoelectric properties in stage II cannot be explained by the ferroelectric phenomenological theory based on the long-range order parameter (spontaneous polarization) for the following reasons. First, according to phenomenological theory, a significant variation of the dielectric/piezoelectric response is generally associated with a long-range phase transition. However, no phase transition occurs in stage II, as confirmed by cryogenic experiments on the long-range crystalline structure (Fig. 2, Supplementary Figs 5 and 6, Supplementary Table 2 and refs. 32,33), the domain patterns on the micron-scale (Supplementary Fig. 7) and the spontaneous polarization (Supplementary Fig. 4). Second, the dielectric relaxation frequency is found to be in the range of $10^0$–$10^8$ Hz (much lower than the typical phonon frequency) at temperatures of 120 and 150 K (Supplementary Figs 8 and 9 and Supplementary Note 1), demonstrating that a considerable portion of the dielectric response is associated with the switching of specific dipoles and/or interface motion. This contribution cannot be simply explained by ferroelectric long-range order. Therefore, it can be concluded that the major difference between relaxor–PT crystals and classical ferroelectrics exists in stage II. In other words, the enhancement of piezoelectricity in stage II can be recognized as a cornerstone for the ultrahigh piezoelectric activity in relaxor–PT crystals when compared with classical ferroelectric crystals, as schematically shown in Fig. 1h. Thus, exploring the mechanism(s) responsible for the dielectric/piezoelectric variation in stage II should enable us to clarify why relaxor–PT crystals exhibit significantly higher dielectric/piezoelectric properties than those of classical ferroelectrics.

**Phase-field simulations**. We use the phase-field method of ferroelectric domain structures[34] to simulate the mesoscale microstructure of relaxor–PT crystals to guide the understanding of the mesoscale mechanism(s) underpinning the high dielectric and piezoelectric properties. Phase-field method has been used to model the effects of random defects/fields on ferroelectric domains and domain-switching to simulate the relaxor behaviour[35–38]. In this work, we consider two important mesoscale characteristics of relaxor–PT: (i) the coexistence of PNRs and long-range ferroelectric domains; and (ii) the symmetry of PNRs being different from that of the long-range ferroelectric domains. For

example, the symmetries of PNRs and long-range ferroelectric domains were found to be orthorhombic and rhombohedral in PZN–xPT ($x < 0.08$) crystals, respectively[39,40]. Thus, relaxor–PT solid solution crystals are treated as PNR-ferroelectric nanocomposites in the phase-field simulations.

In the following simulation, a crystal with orthorhombic PNRs and tetragonal ferroelectric domain matrix is selected as a representative model system. The Curie temperature of the PNRs is assumed to be 620 K based on the Burns temperature of relaxor–PT systems[18], while the Curie temperature of the tetragonal matrix is the same as the Curie temperature of PZN–0.15PT, that is, 468 K. Below the Curie temperature, we assume there are no ferroelectric phase transition for both the orthorhombic PNRs and/or tetragonal matrix. The spontaneous polarization of the orthorhombic PNRs and tetragonal matrix is set to be 0.40 C m$^{-2}$ at 300 K. On the basis of neutron-scattering results[41,42] and the activation energy of PNR switching (Supplementary Fig. 10 and Supplementary Note 2), the diameter of PNRs is assumed to be 3–5 nm with a random distribution, while the volume fraction of PNRs is set to be 7.5%, at which the tetragonal-like domain structure is maintained (Supplementary Fig. 11). Volume fractions in the range of 5–15% yielded similar results as given in Supplementary Fig. 12.

Figure 3 describes the simulated results of a [100]-poled PNR-ferroelectric composite at 100 K. At temperatures below 100 K, PNRs tend to retain their original orthorhombic state dictated by their Landau energy despite the fact that electrostatic, gradient and elastic energies are significant due to the discontinuity of polarization and strain around the interfaces between PNRs and ferroelectric matrix. It can be seen that only PNRs with polar vectors along [110] or [1$\bar{1}$0] exist, while PNRs with polar vectors along [$\bar{1}$10] or [$\bar{1}\bar{1}$0] are absent due to unfavourable electrostatic interaction energy within the tetragonal ferroelectric matrix. The average symmetry of the [100]-poled PNR-ferroelectric composite is still 4mm although there exist PNRs with mm2 symmetry.

With increasing temperature, the polar vectors of some PNRs transform to the [100] direction, that is, the polar vector of PNRs becomes 'collinear' with the polar direction of the ferroelectric matrix, as shown in Fig. 4a. The reason why PNRs transform to the 'collinear' state is that this transformation minimizes the free energy of the entire composite system. In the 'collinear' state, although PNRs' Landau energy is high, the total electrostatic, gradient and elastic energies are low since the discontinuousness of polarization and strain is minimal for the 'collinear' state. As temperature increases, the difference in PNR's Landau energy between orthorhombic and tetragonal states decreases. At relatively high temperatures, therefore, the decrease in the total

electrostatic, gradient and elastic energies can more than offset the increase in Landau energy arising from the transformation of some orthorhombic PNRs to the tetragonal phase. In general, smaller and more isolated PNRs transform to the 'collinear' state at relatively lower temperatures. At temperatures above 350 K, all PNRs are in the 'collinear' state as shown in Fig. 4a.

Figure 4b presents the calculated transverse and longitudinal (small inset) dielectric permittivities of a PNR-ferroelectric composite with respect to temperature. The transverse dielectric permittivity $\varepsilon_{11}/\varepsilon_0$ of the composite is found to be almost the same as that of the matrix for temperatures below 100 K. As the temperature increases from 100 to 200 K, the permittivity $\varepsilon_{11}/\varepsilon_0$ of the composite significantly increases from 1,600 to 4,500, while in the temperature range of 200–400 K, the permittivity of the composite is found to be quite stable with respect to temperature. It should be noted here that the simulated piezo-electric coefficient $d_{15}$ shows a similar temperature dependence to the transverse permittivity $\varepsilon_{11}/\varepsilon_0$, as given in Supplementary Figs 15 and 16. In contrast, the longitudinal dielectric permittivity $\varepsilon_{33}/\varepsilon_0$ of the composite is found to possess the same value as the permittivity of the matrix, indicating that the impact of PNRs on the single-domain longitudinal dielectric response is minimal. The simulated temperature-dependent dielectric responses are in good agreement with experimental observations for PZN–0.15PT crystals (Fig. 1c and Supplementary Fig. 1c).

Let us now discuss the mechanism(s) responsible for the enhanced transverse dielectric permittivity in a PNR-ferroelectric composite. Figure 5a,b display the microstructural evolution and polarization variation of the [100]-poled PNR-ferroelectric composite under a perpendicular [010] E-field, respectively. From Fig. 5a, the enhanced transverse dielectric response can be attributed to two contributions (schematically depicted in Fig. 5c): I—for 'non-collinear' PNRs, they may switch from one stable state to another whose polar direction is favoured by the E-field, that is, PNRs along the [1$\bar{1}$0] direction can be switched to the [110] direction under an [010] E-field (switching may also occur via a metastable state, that is, with the polar vector along the [100] direction); II—for 'collinear' PNRs, their polarizations are much easier to rotate under a perpendicular E-field when compared with the matrix, and this rotation will also facilitate the polarizations of the nearby regions to rotate (please see the

colour variation of the grids near PNRs) to reduce electrostatic, elastic and gradient energies at the interfaces. During this process, the average (macroscopic) polarization rotation under a perpendicular E-field is highly facilitated by PNRs.

At 50 K, both contributions I and II disappear, since PNRs are 'frozen' in the ferroelectric matrix. Therefore, the dielectric response of the composite is similar to that of the ferroelectric matrix. At 150 K, PNRs become unstable and some of them are in the 'collinear' state. Under this condition, both I and II contribute to the enhanced dielectric response. With increasing temperature, more PNRs become 'collinear' and consequently contribution II becomes dominant. At 350 K, all the contribution of PNRs comes from II. On the basis of the simulated transverse polarization-electric field (PE) response of the PNR-ferroelectric composites (Fig. 5b), a relatively high hysteresis is observed at 150 K. This is due to the fact that the switching of PNRs (contribution I) needs to overcome specific energy barriers. At both 50 and 350 K, a linear PE response without hysteresis is observed, since contribution I is absent. The linear PE response at 350 K also indicates that contribution II is not accompanied by hysteresis, which is very important for practical piezoelectric and dielectric applications.

Contribution II raises the question 'why are PNRs easy to rotate when they are in the 'collinear' state?' In the PNR-ferroelectric composite, the transformation of a PNR to a 'collinear' state is driven by the reduction in elastic, gradient and electrical energies of the system. To study the properties of PNRs in the 'collinear' state, it is reasonable to use a [100] d.c. E-field as the driving force to mimic the situation, where PNRs are allowed to rotate to the [100] direction. Figure 6 shows the Landau energy profile for an orthorhombic PNR under various levels of [100] d.c. E-field. At zero d.c. E-field, the polar vector of the stable state is along ⟨110⟩ for an orthorhombic PNR. With increasing d.c. E-field, the polar vector approaches the [100] direction, and the energy barrier among the stable states decreases, as shown in Fig. 6a,b. The lower energy barrier indicates the ease of PNR switching from one stable state to another under an external stimulus. When the driving force is sufficiently high (that is, [100] d.c. E-field ≥180 MV m$^{-1}$), the polar vector of a PNR is rotated to the [100] direction, analogous to the scenario where PNRs are in the 'collinear' state within the ferroelectric matrix. At this condition, as shown in Fig. 6c, the

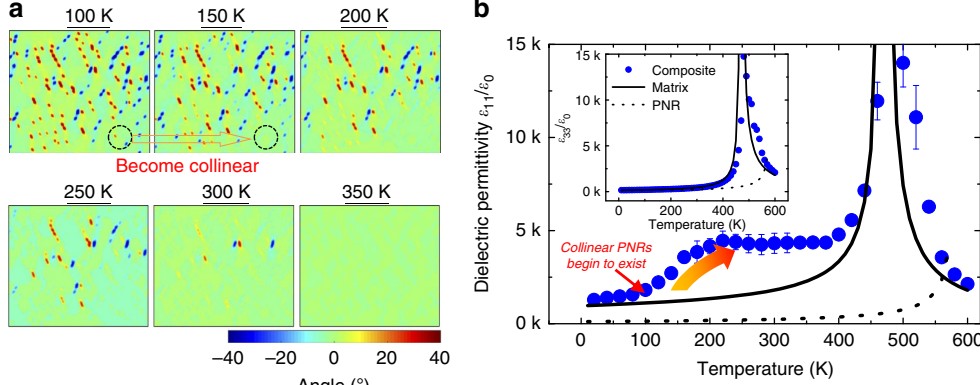

**Figure 4 | Temperature-dependent microstructure and dielectric response for [100]-poled PNR-ferroelectric composites. (a,b)** Temperature-dependent microstructures and dielectric permittivities, respectively. In **a**, the colour bar denotes the angle between the polar vector of the grids and the [100] direction. Here an example of specific PNRs' transformation to the 'collinear' state is given with increasing temperature from 100 to 150 K, as marked by the black dashed circles. In **b**, the intrinsic dielectric permittivities of the ferroelectric matrix and the PNR are given for comparison. The error bars represent the s.d.'s and are obtained from five different random distributions of PNRs with a volume fraction of 7.5%. For calculation of the dielectric permittivities, the magnitude and period of the a.c. E-field were $10^4$ V m$^{-1}$ and $10^5$ time steps, respectively. The corresponding three-dimensional simulations, given in Supplementary Figs 13 and 14, show similar results to the two-dimensional simulations.

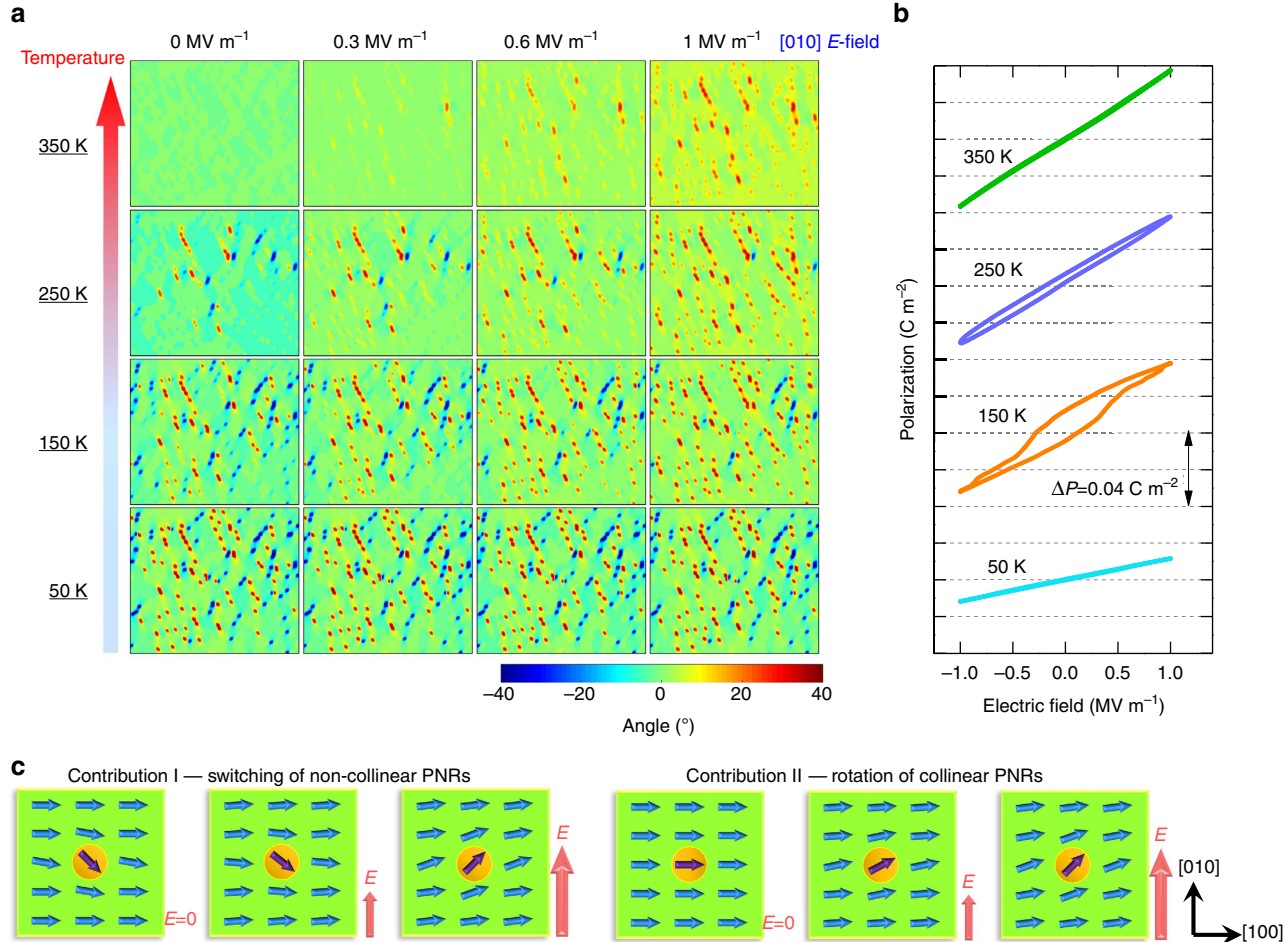

**Figure 5 | Perpendicular *E*-field-induced microstructure and polarization variations for the [100]-poled PNR-ferroelectric composites.**
(**a**) Microstructure evolution for the [100]-poled PNR-ferroelectric composites with applied perpendicular *E*-field along [010] at various temperatures. The colour bar denotes the angle between the polar vector of the grids and the [100] direction. (**b**) Simulated transverse PE responses for the PNR-ferroelectric composite, where the amplitude and period of the a.c. electric field are $1\,MV\,m^{-1}$ and $10^5$ time steps, respectively (see Supplementary Movies 1–3 for details of *E*-field-induced microstructure evolution). (**c**) Schematics for contribution I and contribution II of PNRs to the transverse dielectric response, where the orange regions represent PNRs, and the polar vectors of PNRs and ferroelectric matrix are shown by purple and blue arrows. The corresponding three-dimensional simulation can be found in Supplementary Figs 17 and 18, showing similar results to the two-dimensional simulation.

polarization rotation path is greatly flattened. This flattening explains why 'collinear' PNRs are relatively easy to rotate by a perpendicular *E*-field. It is worth noting that as the d.c. *E*-field further increases, the polar vector of a PNR will be further stabilized along the [100] direction, making the polarization rotation path steeper as presented in Fig. 6d.

According to the above analysis, the relatively stable transverse dielectric permittivity over the temperature range of 200–400 K can also be explained. With increasing temperature, it is expected that the impact of a driving force on PNRs becomes more prominent due to a decrease in the Landau energy barrier between tetragonal and orthorhombic phases. The enhanced impact of the driving force includes the following: (1) stabilizing the 'collinear' PNRs in the matrix (as depicted in Fig. 6d), leading to a decrease in the dielectric response; and (2) destabilization of the 'non-collinear' PNRs, whereupon they transform to the 'collinear' state (as depicted in Fig. 6a–c), leading to an increase of dielectric response. Furthermore, the dielectric response of the ferroelectric matrix increases with increasing temperature. Thus, over the temperature range of 200–400 K, several factors may compete with each other, leading to an overall temperature-stable dielectric response, as simulated in Fig. 4b and experimentally observed in PZN–0.15PT crystals (Fig. 1c).

## Discussion
On the basis of the experimental and phase-field simulated results, the contribution of PNRs to the dielectric and piezo-electric properties in relaxor–PT crystals can be explained on the mesoscale. At room temperature, the high piezoelectric activity of relaxor–PT crystals is mainly associated with the 'collinear' PNRs, and not from the switching of 'non-collinear' PNRs, thus loss and hysteresis are extremely small at room temperature (Supplementary Figs 19–21). With decreasing temperature, the 'collinear' PNRs begin to become isolated from the ferroelectric matrix due to Landau energy dominating the polar direction of PNRs at low temperatures, resulting in a decreased contribution of 'collinear' PNRs, but with an increased contribution from PNR switching. Thus, increased loss and hysteresis associated with PNR switching are observed. By further decreasing temperature, all PNRs are isolated from the matrix and then become frozen, consequently contributions to the dielectric and piezoelectric properties are minimal.

In addition to the explanation of the ultrahigh dielectric/piezoelectric response and related temperature dependence, the phase-field modelling results are also consistent with the microstructural observations of relaxor–PT materials[41,43–45]. Neutron and X-ray diffraction studies have shown that diffuse

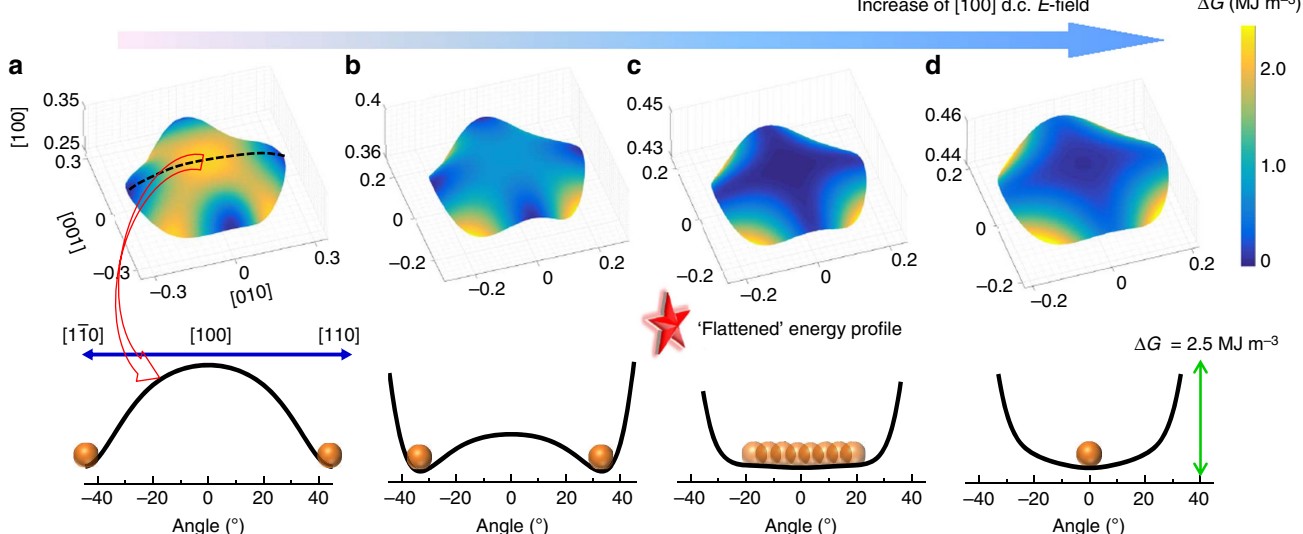

**Figure 6 | The variation of Landau energy profile for an orthorhombic PNR under [100] d.c. *E*-field at 350 K.** (**a**) At the d.c. *E*-field of 0 MV m$^{-1}$; (**b**) at the d.c. *E*-field of 70 MV m$^{-1}$; (**c**) at the d.c. *E*-field of 180 MV m$^{-1}$; (**d**) at the d.c. *E*-field of 250 MV m$^{-1}$. The axes in the three-dimensional figures represent the polarizations along the [100], [010] and [001] directions (unit: C m$^{-2}$). The distance between the point on the figure surface and the origin is the optimized polarization, at which the free energy is minimized for the corresponding direction. The value of energy density is described by the colour and shown in the colour bar, where the minimal energy density is set as the reference, and the scale of energy density is 2.5 MJ m$^{-3}$ for all energy profiles. To clearly show the energy path for polarization rotation, the corresponding two-dimensional (2D) energy profiles for a [1$\bar{1}$0]–[100]–[110] polarization rotation path are given, where the *x* axis represents the angle between the polar vector and the [100] direction, and the *y* axis represents the minimal energy for a specific polar direction. The minimal energy states are represented by the orange balls in the 2D figures.

scattering becomes stronger at lower temperatures[41,43], indicating the local symmetry of PNRs deviates more from the macroscopic symmetry, and/or there is an increased volume proportion of 'non-collinear' PNRs at lower temperatures. This is consistent with the scenario shown in Fig. 4a. In addition, convergent beam electron diffraction patterns obtained by transmission electron microscope revealed that the local symmetry could be lower (monoclinic or triclinic) than the long-range symmetry in PMN–PT crystals[44,45]. This is also confirmed in our simulations where the symmetry of some specific grids (local structure) could be totally broken by the interaction between the ferroelectric matrix and the PNRs, as shown in Fig. 3.

It is worth contrasting our study with previous phonon studies on the contribution of PNRs. Phonon studies showed that the PNR–phonon interaction can induce a phase instability (softening of the transverse acoustic mode) in relaxor–PTs, thus this phenomenon was thought to play an important role in the ultrahigh piezoelectric response[23]. In our model, it is proposed that PNRs facilitate polarization rotation and enhance the shear piezoelectric response of the single domain state, as shown in Fig. 5 and Supplementary Fig. 15. This suggests a possible connection between the softening of the TA mode (shear) and the PNRs.

In summary, based on combined experimental observations and phase-field simulations, the contributions of PNRs to the dielectric and piezoelectric properties of relaxor–PT crystals were quantified, accounting for 50–80% of the room-temperature values. At room temperature, the high piezoelectricity was attributed to the 'collinear' PNRs, which can facilitate polarization rotation and enhance the shear piezoelectric response of a single-domain state. The impact of such local structure on the macroscopic properties is not limited to relaxor–PT systems. It also exists in other material systems with structural fluctuations[46–48], such as lead barium niobate $Pb_{1-x}Ba_xNb_2O_6$ (ref. 47) and cubic pyrochlores $Bi_{1.5}Zn_{1.0}Nb_{1.5}O_7$ (ref. 48). Therefore, local structural engineering has the potential to be an effective and

general method for the future design of high-performance functional materials.

## Methods

**Sample preparation.** The PMN–0.28PT and PMN–0.32PT single crystals were grown by a modified Bridgman method. The PZN–0.15PT crystals were grown by high-temperature flux method. The samples were oriented by real-time Laue X-ray back diffraction. Vacuum-sputtered gold electrodes were applied to the faces of the samples. The PMN–0.28PT, PMN–0.32PT and PZN–0.15PT single crystals were poled along their respective polar directions, that is, [111], [011] and [001]. To obtain single-domain states, a high-temperature poling approach was used[49]. For the PMN–0.28PT and PMN–0.32PT crystals, the samples were poled at 100 °C with a 1 MV m$^{-1}$ *E*-field, and then cooled to room temperature with the *E*-field kept on. For the PZN–0.15PT crystals, the samples were poled at 250 °C with a 0.2 MV m$^{-1}$ *E*-field, subsequently cooled to room temperature under *E*-field. After poling, the electrodes were removed and re-electroded on the (1$\bar{1}$0), (100) and (100) faces for the PMN–0.28PT, PMN–0.32PT and PZN–0.15PT crystals, respectively, for allowing determination of the transverse dielectric permittivities and shear piezoelectric coefficients. The single-domain dielectric and piezoelectric coefficients were measured using the standard coordinate systems of single-domain crystals. For rhombohedral, orthorhombic and tetragonal single-domain crystals, the standard coordinate systems are $X$: [1$\bar{1}$0] × $Y$: [11$\bar{2}$] × $Z$: [111], $X$: [0$\bar{1}$1] × $Y$: [100] × $Z$: [011] and $X$: [100] × $Y$: [010] × $Z$: [001], respectively.

**Characterization.** Dielectric measurements. The temperature dependence of the various dielectric permittivities was measured using an LCR meter (HP4980) being connected to a computer-controlled temperature chamber (high temperature: home-made temperature controller; low temperature: Model 325 Cryogenic Temperature Controller, Lake Shore). Wide-frequency spectra ($10^{-1}$–$10^5$ Hz) of dielectric response were measured at various temperatures by High Performance Modular Measurement System (Novocontrol Technologies).

Piezoelectric measurements. Shear piezoelectric coefficients were determined by the resonance method, following the IEEE standard on piezoelectricity, using an HP4294 impedance analyser.

X-ray diffraction measurements. X-ray analysis was carried out using an X-ray diffractometer (PANalytical) with Cu Kα radiation with temperature controller. The operation voltage and current were 40 kV and 40 mA, respectively.

Synchrotron X-ray diffraction measurements. *In situ* synchrotron X-ray diffraction experiments were performed at Argonne National Laboratory beam line 11-BM from 300 down to 100 K. The wavelength is 0.414212 Å. The instrument resolution is $\Delta d/d = 0.00017$, representing the state-of-the-art *d*-spacing resolution for diffraction measurements.

**Spontaneous polarization.** The room-temperature spontaneous polarization was determined by the polarization-electric field loops, which were measured using a modified Sawyer–Tower circuit. To determine the variation of spontaneous polarization with respect to temperature, the pyroelectric current was measured using a KEITHLEY6485 picoammeter connected to a temperature chamber (Delta Design 2300).

**Phase-field simulation.** The temporal evolution of the polarization field and the domain structure evolution are described by the time-dependent Ginzburg–Landau equations

$$\frac{\partial P_i(\mathbf{r},t)}{\partial t} = -L\frac{\delta F}{\delta P_i(\mathbf{r},t)}, \quad (i=1,2,3),\tag{1}$$

where $L$ is the kinetic coefficient, $t$ the time, $F$ the total free energy of the system and $P_i(\mathbf{r},t)$ the polarization. The total free energy of the system includes the bulk free energy, the elastic energy, the electrostatic energy and the gradient energy:

$$F = \int_V \left[f_{bulk} + f_{elas} + f_{elec} + f_{grad}\right] dV\tag{2}$$

where $V$ is the system volume of the PNR-ferroelectric composite, $f_{bulk}$ denotes the bulk free energy density, $f_{elas}$ the elastic energy density, $f_{elec}$ the electrostatic energy density and $f_{grad}$ the gradient energy density. The detailed expressions and parameters of these energy densities are given in Supplementary Note 3.

The impacts of volume fraction of PNRs (Supplementary Fig. 12 and Supplementary Note 4) and thermal noise (Supplementary Figs 22–25 and Supplementary Note 5) were also included and discussed in the phase-field simulations.

In the computer simulations, we used two-dimensional $128 \times 128$ discrete grid points (three-dimensional: $64 \times 64 \times 64$ grid points) and periodic boundary conditions. The grid space in real space is chosen to be $\Delta x = \Delta y = 1$ nm.

**Data availability.** The data corresponding to this study are available from the first author and corresponding authors on request.

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

## Acknowledgements

This work was supported by ONR (Grants Nos. N00014-12-1-1043 and N00014-12-1-1045). F.L. acknowledges the support by the Office of China Postdoctoral Council, the National Natural Science Foundation of China (Grant Nos. 51572214 and 51372196), the Natural Science Foundation of Shaanxi province (2015JQ5135) and the 111 Project (B14040). S.Z. thanks to the support of ONRG (Grant No. N62909-16-1-2126). T.Y. acknowledges the support of National Science Foundation under the Grant Number DMR-1410714. L.-Q.C. is supported by U.S. Department of Energy, Office of Basic Energy Sciences, Division of Materials Sciences and Engineering under Award DE-FG02-07ER46417. F.L. and G.L. thank Dr Saul Lapidus for the technical supports of synchrotron X-ray diffraction experiments. J.L.W. and Z.C. thank Qingyong Ren for the cryogenic X-ray diffraction measurements. The use of the Advanced Photon Source was supported by the U.S. Department of Energy, Office of Science and Office of Basic Energy Sciences, under Contract No. DE-AC02-06CH11357.

## Author contributions

The project was conceived and designed by F.L., S.Z., Z.X., T.R.S. and L.-Q.C.; F.L. prepared crystal samples, performed the dielectric and piezoelectric experiments, and the phase-field simulations; T.Y. and J.J.W. assisted in phase-field simulations; Z.-G.Y. assisted in the analysis of experimental dielectric responses and discussed with F.L. on the concepts of PNRs; G.L. performed synchrotron X-ray diffraction experiments on powders; N.Z. did the Rietveld refinements of the structures. J.L.W. and Z.C. performed the cryogenic X-ray diffraction experiments for crystals; J.L. and S.Z. grew the crystals; F.L., S.Z., Z.-G.Y., T.R.S. and L.-Q.C. prepared the manuscript.

## Additional information

**Competing financial interests:** The authors declare no competing financial interests.

