## [Peer Review File · Nature Communications]

Reviewers' Comments:

Reviewer #1 (Remarks to the Author):

This paper attempts to explain the ultrahigh piezoelectric response of relaxor-based ferroelectrics in terms polar nanoregions (PNRs). It is widely assumed that PNRs play a role in enhancing the piezoelectric response in relaxor-based ferroelectrics, but the mechanism has never been fully explained. This paper tries to explain the mechanism. This is certainly an important objective. However, I cannot understand the argument and it does not seem applicable in the most technologically important case. I am not convinced for the following reasons:

(1) The explanation for the PNR contribution to the ultrahigh piezoelectric response in this paper is based on the co-alignment of the PNRs (in field) with the macroscopic ferroelectric domains (from page 6, "...the polar vector of PNR is collinear with the polar direction of the ferroelectric matrix, so-called collinear state."). However, the ultrahigh piezoelectric response used in applications is based on the domain engineered 4R structure in rhombohedral PMN-PT, in which the crystals are poled along [100] and the macro ferroelectric domains align along the 4 closest [111]-type rhombohedral directions (see, for example, Zhang et al. J. Appl. Phys. 111, 031301 (2012)). Based on diffuse scattering experiments the PNRs have components of the local displacements along [100] and [110] even though the macro domains do not (see, for example, Xu et al. PRB 82, 134124 (2010)). In other words, the highest performing relaxor-based ferroelectrics are configured in a way that is different from what would be expected from the co-alignment mechanism described in this paper. If the poling field is applied along [100] and the small PNRs align along this direction, as suggested by the model, they would not be co-aligned with the 4R macro domains pointing along [111] directions. So PNR co-alignment with domains could not be driving the enhanced performance in this most important case.

(2) I also don't understand the description given in the abstract that the collinear PNRs act as "seeds" for the polarization rotation. The polarization rotations involve a shear deformation on a macroscopic scale. The expression "seeds" sound like a local nucleation process. How do the PNRs on the nanoscale facilitate a macroscopic domain rotation? Do they shear more easily than the matrix? There are composites with soft components that are nonetheless macroscopically stiff. So this does not seem obvious to me. And why do they have to be collinear with the ferroelectric matrix in the first place? This is not explained well.

(3) A smaller issue that is not explained properly in the introduction is that a part of the increased performance of relaxor-based ferroelectrics over PZT-based ferroelectrics is that they can be grown as single crystals. Comparing the single crystal performance with the polycrystalline ceramic is a little misleading. I agree that the PNRs (relaxor component) probably benefits performance, but this statement should be qualified by the fact that single crystals are being compared with a polycrystalline ceramic.

(4) The rise in the piezoelectricity and dielectric permittivity at low temperatures (stage II) is consistent with a process becoming thermally activated. While this could be related to PNR rotations, the appearance looks generic. The activation appears to initiate at close to $T = 0$ K, and top off at 100 K to 200 K and then saturate. This looks a little like the population of thermal vibrations, which can manifest in related properties in the same way. For example, the thermal expansion coefficient is well known to track the heat capacity in this way because of the way phonons contributing to thermal expansion become populated. The temperature range of 100 K to 200 K is consistent with ~ 10 meV to ~ 20 meV excitations, which is the right scale for the low energy TO and TA phonon modes associated with ferroelectric behavior. Of course, the dynamics of the PNRs would be similar. The point is that the model is not uniquely constrained by the data. Many things become thermally excited in this temperature range.

(5) Early on they mention a relationship between the PNRs and a softening of the transverse

acoustic phonons in Ref. [21]. However, they never explain this known relationship. A softening of the shear mode relates directly to the macroscopic polarization rotation, and hence the ultrahigh piezoelectric response. In Ref. [21] they show that an alignment of the PNRs along [111] softens the [110]-TA phonon. The softening of this phonon at both long and short wavelengths implies a modification of the shear stiffness across multiple length scales. Can the model explain this behavior?

Reviewer #2 (Remarks to the Author):

The authors present compelling dielectric and piezoelectric data that suggests that the strong enhancement of the transverse permittivity on heating below ~ 200 K is a prerequisite for the ultrahigh piezoelectricity observed in relaxor ferroelectrics. The authors support their idea with data measured on three different, single-domain, relaxor systems: PMN-0.28PT, PMN-0.32PT, and PZN-0.15PT.

The low-temperature enhancement of the transverse dielectric permittivity is, in each case, accompanied by a significant frequency dispersion, suggesting that the relaxor character is essential. As polar nano-regions (PNR) are widely viewed as underlying relaxor behavior, the conclusion is drawn that the PNR are responsible for the enhancement, and thus the bulk of the ultrahigh piezoelectricity. The authors further support their claim using phase field simulations.

In general, I am favorably inclined to recommend this manuscript for publication. Relaxors, and in particular MPB compositions, are of extremely high interest within the physics and materials science communities. However, before doing so I have several questions:

(1) PZN-0.15PT is a composition that lies on the Ti-rich/tetragonal side of the MPB. Thus I would not have expected it to exhibit relaxor character. The presence of PNR is most directly manifested by the appearance of x-ray or neutron diffuse scattering, which reflects local/short-range structural correlations. As shown by Stock et al., Phys. Rev. B 73 064107 (2006), and Matsuura et al., Phys. Rev. B 74, 144107 (2006), this diffuse scattering vanishes for PMN-xPT compositions on the tetragonal side of the MPB. Can the authors offer a way to reconcile their findings with these two studies?

(2) If "the impact of PNRs on the longitudinal dielectric response is minimal", then what mechanism is responsible for the very large values of d_{33} reported by Guo et al, J. Phys. C 15, L77 (2003) (Fig. 5) for [001] and [110]-oriented single crystals of PMN-xPT near the MPB? Indeed, the data of Guo et al. seem to contradict the authors statement on page 3 that: "... the transverse dielectric and shear piezoelectric responses are significantly larger than their longitudinal counterparts, and thus are the dominant factors for the high performance of relaxor-PT crystals [25-27]." Please comment on this.

Other than this, there are minor grammatical mistakes throughout the paper that should be corrected by a native speaker of English.

Reviewer #3 (Remarks to the Author):

In this manuscript the authors attempt to disentangle the question of the "Origin of ultrahigh piezoelectric responses in relaxor-based ferroelectrics" and proceed quite successfully on the basis of a phenomenological description centrally involving the activity of polar nanoregions (PNR) in several relaxor crystals of PMN-PT and PZN-PT. The authors' final statement "... the contribution of PNRs to the dielectric and piezoelectric response in relaxor-PT crystals can be clearly elucidated," sounds very optimistic and seems to denote a breakthrough.

Unfortunately this promise does not hold at a closer look. Although the keyword "PNR" has 93 times quoted, the origin of these fancy "polar nanoregions" has not at all been physically explained or theoretically deduced. Instead, the very understanding of the physics behind the PNR has remained highly doubtful. PNR are simply characterized as "nanoscale inhomogeneities" with "diameters of 3-5 nm, random distribution and volume fraction $\sim 7.5\%$ ". In order to stress their apparent segregation from the ferroelectric "host" crystal a novel and uncommon designation of "PNR-ferroelectric composites" has been chosen to replace the long-accepted term "relaxor ferroelectric". Obviously PNR is considered synonymous to an unknown "defect" and reminds of the very early time of semiconductor physics, where impure materials like Ge and Si showed unexpected phenomena, which were to be understood only many years later.

Actually, however, the authors have neglected recent progress of relaxor physics:

1. I. K. Jeong et al. have found (Phys. Rev. Lett. 94 (2005) 147602) that "the volume fraction of the PNR [in PMN] as a function of temperature increases from 0% to a maximum of $\sim 30\%$ as the temperature decreases from 650 to 15 K. Below $T \sim 200$ K the volume fraction of the PNRs becomes significant, and PNRs freeze into the spin-glass-like state". Thus PNR take clearly part in the intrinsic thermodynamics of the relaxor crystal and thus influence many of its properties. In particular, the "spin glass-like state" of the PNR was recently evidenced by W. Kleemann at their percolation limits in SBN and BTZ (Phys. Stat. Sol. (b) 251 (2014) 1993). Relaxors have thus joined the family of "ferroic glasses" like strain and spin cluster glass (X. B. Ren, Phys. Stat. Sol. (b) 251 (2014) 1982). According to Jeong et al. (2005) PNR percolation coincides with the superglass transition in PMN at $T_g \sim 239$ K (W. Kleemann, J. Dec, unpublished).
2. D. Phelan, Z.G. Ye (!), P.M. Gehring et al. (PNAS 111 (2014) 1754) stressed the "Role of random electric fields (RFs) in relaxors" to be "implicated as the genesis of relaxor behavior." Hence, the authors' paragraph on p. 10 claiming "the presence of RFs cannot explain the high transverse dielectric and shear piezoelectric response in the relaxor-PT crystals" must be cast severely in doubt. First of all, the RF-assisted creation of static PNR below the Burns temperature $T_d \sim 600$ K has escaped the authors' modeling. Instead, PNR were taken as defects like dopants in a semiconductor. Thus they missed the outcome of the intrinsic disorder of heterovalent cations, which creates a "frozen" charge disorder and, hence, random electric fields with many unusual consequences, viz. the essence of the "enigmatic relaxor physics"
3. B.-X. Xu, S. Wang, M. Yi (Proc. Appl. Math. Mech. 15 (2015) 723/ DOI 10.1002/pamm.201510348) have considered "A finite element phase field model for relaxor ferroelectrics". The model is derived from thermodynamic analysis including the material force theory. Random field theory is adopted to take the disorder of relaxor ferroelectrics into account. Results show that the model is capable of reproducing relaxor features, such as domain miniaturization, small remnant polarization and large piezoelectric response. Dependence of these features on the random field strength is in line with experimental experience. Since the present authors' incomplete attempts of a relaxor phase field theory cannot compete with the last-cited professional one, I see no chance of publication.

In view of these large deficiencies the manuscript is not acceptable for NComms.

Dear referees,

Thanks very much for your helpful comments and suggestions. We have revised our manuscript accordingly, and the point-by-point responses to comments are enclosed in this letter.

Response to referee #1

Comment: This paper attempts to explain the ultrahigh piezoelectric response of relaxor-based ferroelectrics in terms polar nanoregions (PNRs). It is widely assumed that PNRs play a role in enhancing the piezoelectric response in relaxor-based ferroelectrics, but the mechanism has never been fully explained. This paper tries to explain the mechanism. This is certainly an important objective.

Reply: We thank the reviewer for his or her appreciation for the importance of our work.

Comment: However, I cannot understand the argument and it does not seem applicable in the most technologically important case. I am not convinced for the following reasons:

(1) The explanation for the PNR contribution to the ultrahigh piezoelectric response in this paper is based on the co-alignment of the PNRs (in field) with the macroscopic ferroelectric domains (from page 6, "...the polar vector of PNR is collinear with the polar direction of the ferroelectric matrix, so-called collinear state."). However, the ultrahigh piezoelectric response used in applications is based on the domain engineered 4R structure in rhombohedral PMN-PT, in which the crystals are poled along [100] and the macro ferroelectric domains align along the 4 closest [111]-type rhombohedral directions (see, for example, Zhang et al. J. Appl. Phys. 111, 031301 (2012)). Based on diffuse scattering experiments, the PNRs have components of the local displacements along [100] and [110] even though the macro domains do not (see, for example, Xu et al. PRB 82, 134124 (2010)). In other words, the highest performing relaxor-based ferroelectrics are configured in a way that is different from what would be expected from the co-alignment mechanism described in this paper. If the poling field is applied along [100] and the small PNRs align along this direction, as suggested by the model, they would not be co-aligned with the 4R macro domains pointing along [111] directions. So PNR co-alignment with domains could not be driving the enhanced performance in this most important case.

Reply: We truly appreciate the detailed and concrete comments by the reviewer, so we are able to respond accordingly. Our proposed mechanism is directly applicable to the technologically important case as explained below, and it is our fault that we did not make it clear in the original manuscript. We have revised the manuscript in the introduction to make this connection.

The reviewer is correct that the optimum longitudinal piezoelectric coefficient d_{33}^* and dielectric permittivity ϵ_{33}^* are indeed generally observed in relaxor-PT crystals poled along the nonpolar directions. Based on previous investigations [Damjanovic D *et al*, *Appl. Phys. Lett.* **83**, 527-529 (2003); Li F *et al*, *Advanced Functional Materials* **21**, 2118 (2011) and references therein], this characteristic is attributed to the very high shear piezoelectric response and transverse dielectric response of single domain relaxor-PT crystals (Table S1 in Supplementary information), giving rise to the highest d_{33}^* and ϵ_{33}^* values being deviated from the spontaneous polar direction, as shown in

Fig. R1. Thus, the high longitudinal dielectric permittivity ϵ_{33}^* and piezoelectric coefficient d_{33}^* in domain engineered crystals actually originate from the single domain transverse dielectric permittivity ϵ_{11} and shear piezoelectric coefficient d_{15} , respectively. This is the reason why we focus our attention on the shear mode of single domain crystals. Fig. R2 shows the temperature- and frequency-dependent behavior of dielectric permittivity ϵ_{33}^* for [011]-poled PZN-0.15PT and [001]-poled PMN-0.30PT (a rhombohedral crystal and similar to PMN-0.28PT) domain engineered crystals. We can see that the temperature-dependent dielectric permittivity ϵ_{33}^* for domain engineered crystals are similar to that of dielectric permittivity ϵ_{11} in single domain crystals (see Figs. 1 f and d of the main paper). The following sentence was added to the revised paper.

“It should be noted that the large longitudinal piezoelectric properties in domain-engineered relaxor-PT crystals originates from the high shear piezoelectric response in the corresponding single domain state²⁷⁻²⁹.”

Fig. R1 Orientation dependence of piezoelectric coefficient d_{33}^* for (a) rhombohedral, (b) orthorhombic, and (c) tetragonal PIN-PMN-PT crystals. [Li F *et al*, *Advanced Functional Materials* **21**, 2118 (2011)].

Fig. R2 Temperature dependence of longitudinal dielectric permittivity ϵ_{33}^* for (a) a [011]-poled tetragonal PZN-0.15PT crystal and (b) a [001]-poled rhombohedral PMN-0.30PT domain engineered crystal. (Unpublished data)

The reviewer is concerned that PNRs would align along the poling direction for domain-engineered crystals (i.e., crystals poled along the nonpolar directions). Based on our phase-field simulations, we observed that the polar directions of PNRs, after poling, are still controlled by their Landau potential and the local fields (i.e., local electric field, elastic field, and gradient driving field) from

their nearby ferroelectric matrix. As we will show below, PNRs do contribute to the longitudinal dielectric and piezoelectric responses in domain-engineered crystals, based on the same mechanism proposed in our paper.

To clarify this, we performed phase-field simulation for a $[110]$ -poled PNR-ferroelectric composite, as shown in Fig. R3. It can be seen that two different macro-domains with the polar directions pointing to the $[100]$ and $[010]$ directions remain after poling the crystal along the $[110]$ direction. At a high temperature (350 K), most of the PNRs are in the “collinear” state with their respective macro-domains. With decreasing temperature, PNRs are gradually isolated from the matrix. As shown in Fig. R3, the PNRs with polar vectors approaching $[110]$ and $[\bar{1}\bar{1}0]$ directions are present in $[100]$ macro-domains (blue), while in $[010]$ macro-domains (orange), the PNRs with polar vectors approaching $[110]$ and $[\bar{1}\bar{1}0]$ directions are present. It should be noted here that the angles between the polar vectors of PNRs and the macro polarizations are based on the competition of the Landau energy of the PNRs and the local interactions between PNRs and ferroelectric matrix, as a function of temperature.

Fig. R3 Microstructure of $[110]$ -poled PNR-ferroelectric composites (poled at room temperature, i.e., 300 K) from 50 K to 350 K, where the x- and y-axes represent the $[100]$ and $[010]$ directions respectively. The scale of this simulation is 512×512 nm. All the parameters in simulation are the same as the simulations in the main text. The color bar denotes the angle (unit: degree) between the polar vector and the $[100]$ direction. Selected orientations are given in the color bar. The average polar directions of the macro-domains are represented by yellow and light green arrows. (Unpublished data)

Fig. R4 Microstructure variation of [110]-poled PNR-ferroelectric composites under an [110] E-field at 350 K. The color bar denotes the angle (unit: degree) between the polar vector and the [100] direction. The selected orientations are given in color bar. (Unpublished data)

Fig. R5 Simulated temperature-dependent longitudinal dielectric permittivity $\epsilon_{33}^*/\epsilon_0$ for the [110] poled PNR-ferroelectric composite. For calculating the dielectric permittivities, the magnitude and period of the ac E-field were $10^4 \text{ V}\cdot\text{m}^{-1}$ and 10^5 time steps, respectively. (Unpublished data)

Fig. R4 shows the simulated microstructural variation of the [110]-poled PNR-ferroelectric composite with respect to [110] E-field at 350 K. It can be seen that PNRs are more prone to rotation (approaching [011] direction) under the [110] E-field compared to the matrix, which will significantly contribute to the corresponding dielectric and piezoelectric responses, as discussed in the main text. Fig. R5 gives the simulated temperature dependence of dielectric permittivity ϵ_{33}^* of the [110]-poled PNR-ferroelectric composite. Compared with the tetragonal matrix, the large dielectric enhancement in the PNR-ferroelectric composite is due to the contribution of PNRs.

As discussed above, therefore, the mechanisms proposed in our paper are also valid for domain engineered crystals. PNRs are believed to significantly contribute to the longitudinal dielectric and piezoelectric responses in the domain-engineered crystals.

Finally, we would like to clarify that our model is not in contradiction to the diffuse scattering experiments. The major finding in the diffuse scattering experiment [Xu et al. PRB 82, 134124 (2010)] is that PNRs have polar components along the directions other than the polar directions of macro-domains. It can be seen from Fig. R3 that some PNRs are not in “collinear” state even at quite high temperatures (250 K or 350 K). Those PNRs have polar components along the directions

other than the polar directions of macro-domains.

Comment: (2) I also don't understand the description given in the abstract that the collinear PNRs act as "seeds" for the polarization rotation. The polarization rotations involve a shear deformation on a macroscopic scale. The expression "seeds" sound like a local nucleation process. How do the PNRs on the nanoscale facilitate a macroscopic domain rotation? Do they shear more easily than the matrix? There are composites with soft components that are nonetheless macroscopically stiff. So this does not seem obvious to me. And why do they have to be collinear with the ferroelectric matrix in the first place? This is not explained well.

(i) The expression "seeds" sound like a local nucleation process. How do the PNRs on the nanoscale facilitate a macroscopic domain rotation?

Reply: Thanks for the comment. To clarify this issue, we compared the perpendicular E-field induced variation of the local structure (Fig. R6) and the average polarization (Fig. R7), for PNR-ferroelectric composites and ferroelectric matrix, respectively. As shown in Fig. R6 (at 350K), PNRs are much easier to be rotated under a perpendicular E-field when compared to the matrix due to the fact that the polarization rotation path of "collinear" PNRs is greatly flattened due to the impacts of local fields (i.e., local electric field, elastic field, and gradient driving field) on PNRs (shown in Fig. 6 of the main paper). In addition, the rotation of PNRs will induce the nearby lattices to rotate (please see the color variation of nearby regions) for reducing the electrostatic, elastic and gradient energies. Thus, the average (macroscopic) polarization variation (or rotation) under a perpendicular E-field is highly enhanced by PNRs, as shown in Fig. R7. By applying E-field, we can see that the PNRs gradually "isolate" (polar vector of PNRs deviates from that of matrix) from the matrix. It somewhat looks like a nucleation process, so we use the expression of "seeds".

In the revised paper, we deleted the word "seeds" and changed the corresponding expression to avoid misunderstanding, meanwhile we emphasize that the average (macroscopic) polarization variation under a perpendicular E-field can be enhanced by the presence of PNRs.

Fig. R6 Perpendicular E-field ([010] E-field) induced microstructure and polarization variation for the [100]-poled PNR-ferroelectric composites and [100]-poled ferroelectric matrix at 350 K. The

color bar denotes the angle between the polar vector of the grids and the [100] direction. The scale of this simulation is 128×128 nm.

Fig. R7 Simulated transverse polarization-electric field (PE) responses for a [100]-poled PNR-ferroelectric composite and a [100]-poled tetragonal matrix at 350 K. The ac electric field is applied along [010] direction, being perpendicular to the poling direction. The amplitude and period of the ac electric field are 1 MV·m⁻¹ and 10⁵ time steps, respectively.

Comment: (ii) Do they shear more easily than the matrix? There are composites with soft components that are nonetheless macroscopically stiff.

Reply: Thanks for the comment. Yes, PNRs shear more easily than the matrix due to the easier rotation for the polarization of PNR. In perovskite ferroelectrics, an easier polarization rotation corresponds to a higher shear piezoelectric response. The reason why PNRs are easier to be rotated is due to the fact that the polarization rotation path of “collinear” PNRs is greatly flattened due to the impacts of local fields (i.e., local electric field, elastic field, and gradient driving field) on PNRs, as analyzed in Fig. 6 of the main paper. It should be noted here that the easier shearing of PNRs is not their intrinsic nature. Without the impacts of the local fields, PNRs’ rotation is not easy and the corresponding dielectric permittivity is very low (as shown in the dotted black line in Fig. 4 of the main paper).

In order to further clarify this issue (“There are composites with soft components that are nonetheless macroscopically stiff”), we give the simulated results for a PNR-ferroelectric composite with PNR’s size of 10 nm. As shown in Fig. R8, the contribution of PNRs (at 150 K-350 K) significantly decreases if the size of PNRs is increased to 10 nm. This is because that the average local fields on each grid of PNRs are weak in the case of large-size PNRs. As shown in Fig. R9, PNRs are stable and do not transform to the “collinear” state even at 350K as the size is set to be 10 nm. From this example, we can see that “a strong interaction” between PNRs and ferroelectric matrix is essential for inducing the “instability” of PNRs (i.e., a flattened polarization rotation path) and the high dielectric/piezoelectric properties in PNR-ferroelectric composites. Therefore, just adding soft components to a composite may not change its stiffness, in our case it is the *interaction* (competition of the Landau potential of PNRs and the local electrostatic, elastic, and gradient energies) between different components playing the important role in the enhanced transverse dielectric and shear piezoelectric responses.

Fig. R8 Simulated temperature-dependent transverse dielectric permittivity for the [100]-poled PNR-ferroelectric composites under different sizes of PNRs. The volume fractions of PNRs are same, i.e., 7.5%, for two different cases. For calculation of the dielectric permittivities, the magnitude and period of the ac E-field were $10^4 \text{ V}\cdot\text{m}^{-1}$ and 10^5 time steps, respectively.

Fig. R9 Microstructures for the [100]-poled PNR-ferroelectric composites at 150 K and 350 K. The size of PNRs is set to be 10 nm with a random distribution. The volume fraction of PNRs is 7.5%. The color bar denotes the angle between the polar vector of the grids and the [100] direction. The scale of this simulation is $128 \times 128 \text{ nm}$.

Comment: (iii) And why do they have to be collinear with the ferroelectric matrix in the first place? This is not explained well.

Reply: Thanks for the comment, and sorry we didn't explain this issue clearly in the paper. Please check the following explanation and we also added this explanation in the revised paper.

At low temperature, PNRs tend to retain their original orthorhombic state dictated by their Landau energy despite the fact that electrostatic, gradient, and elastic energies are significant due to the discontinuity of polarization and strain around the interfaces between PNRs and ferroelectric matrix. With increasing temperature, the polar vectors of some PNRs transform to the [100] direction, i.e., the polar vector of PNR is "collinear" with the polar direction of the ferroelectric matrix. The reason why PNRs transform to the "collinear" state is that this transformation minimizes the free energy of

the entire composite system. In the “collinear” state, although PNRs’ Landau energy is high, the total electrostatic, gradient, and elastic energies are low since the discontinuousness of polarization and strain is minimal for the “collinear” state. As the temperature increases, the difference in PNR’s Landau energy between orthorhombic and tetragonal states decreases. At relatively high-temperature, therefore, the decrease in the total electrostatic, gradient, and elastic energies can more than offset the increase in Landau energy arising from the transformation of some orthorhombic PNRs to the tetragonal phase.

Comment: (3) A smaller issue that is not explained properly in the introduction is that a part of the increased performance of relaxor-based ferroelectrics over PZT-based ferroelectrics is that they can be grown as single crystals. Comparing the single crystal performance with the polycrystalline ceramic is a little misleading. I agree that the PNRs (relaxor component) probably benefits performance, but this statement should be qualified by the fact that single crystals are being compared with a polycrystalline ceramic.

Reply: Thanks for this valuable suggestion. We agree with the referee. We decided to delete the following sentence: “They dramatically outperform the widely used PbZrO₃-PbTiO₃ (PZT)-based MPB ceramics...”. Instead, we provided the piezoelectric coefficient d_{33}^* (500~1000 pC/N) of PZT crystals measured at near MPB compositions in the revised paper by one of the coauthors (Z.G. Ye). Our previous phase-field simulations showed that the maximum d_{33}^* of a single-crystal PZT near the MPB composition is around 750 pC/N [Y. Cao, G. Sheng, J. X. Zhang, S. Choudhury, Y. L. Li, C. Randall, and L. Q. Chen, Piezoelectric response of single-crystal PbZr_{1-x}Ti_xO₃ near morphotropic phase boundary predicted by phase-field simulation, Appl Phys Lett, 97, 252904 (2010)]

Comment: (4) The rise in the piezoelectricity and dielectric permittivity at low temperatures (stage II) is consistent with a process becoming thermally activated. While this could be related to PNR rotations, the appearance looks generic. The activation appears to initiate at close to T = 0 K, and top off at 100 K to 200 K and then saturate. This looks a little like the population of thermal vibrations, which can manifest in related properties in the same way. For example, the thermal expansion coefficient is well known to track the heat capacity in this way because of the way phonons contributing to thermal expansion become populated. The temperature range of 100 K to 200 K is consistent with ~ 10 meV to ~ 20 meV excitations, which is the right scale for the low energy TO and TA phonon modes associated with ferroelectric behavior. Of course, the dynamics of the PNRs would be similar. The point is that the model is not uniquely constrained by the data. Many things become thermally excited in this temperature range.

Reply: Thanks for the comment. Yes, the dielectric relaxation behavior is quite like a thermal activation process. We analyzed this process by Arrhenius law (section 1.6 in Supplementary information), where the activation energies of PMN-0.28PT, PMN-0.32PT and PZN-0.15PT were found to be around 2000·k_B, i.e., ~170 meV. Due to the following three reasons, we believed the dielectric relaxation behavior in 20 K~200 K was associated with the PNRs. (1) Based on synchrotron XRD experiments, we didn’t observe any evidence indicating a ferroelectric phase transition over the temperature range of 50-300 K, so the variation of dielectric property in this

temperature range was not related to a long-range phase transition. (2) According to the analysis of dielectric spectra (Section 1.5 in Supplementary information), dielectric relaxation frequency was found to be in the range of 10^0 - 10^8 Hz at temperatures of 120 K and 150 K. This frequency range is generally thought to be associated with the switching of specific dipoles and/or interfaces motion. (3) As well, we measured the low-temperature dielectric property for several classical ferroelectrics, such as $\text{Pb}(\text{Zr}_{0.55}\text{Ti}_{0.45})\text{O}_3$ ceramics and $(\text{K}_{0.5}\text{Na}_{0.5})\text{NbO}_3$ crystals (see Fig. R10). The low-temperature dielectric relaxation is not present. Therefore, this dielectric relaxation is thought to be related to a unique structural characteristic of relaxor-PT crystals, i.e., the presence of PNRs.

Fig. R10 Low-temperature dielectric property of (a) a [001]-oriented $(\text{K}_{0.5}\text{Na}_{0.5})\text{NbO}_3$ crystal and (b) a $\text{Pb}(\text{Zr}_{0.55}\text{Ti}_{0.45})\text{O}_3$ ceramic. For $(\text{K}_{0.5}\text{Na}_{0.5})\text{NbO}_3$ crystal, a rhombohedral-orthorhombic (R-O) phase transition is present at 155 K. As shown in this figure, the low-temperature dielectric relaxation is not present in $\text{Pb}(\text{Zr}_{0.45}\text{Ti}_{0.55})\text{O}_3$ and $(\text{K}_{0.5}\text{Na}_{0.5})\text{NbO}_3$, since they do not possess PNRs. (Unpublished data)

Based on the experimental observations about long-range ferroelectric phase and local PNRs' phase (XRD and diffuse scattering experiments), we used the phase-field method to simulate the piezoelectric properties of relaxor-PT crystals on the mesoscale. According to the simulation work, we proposed a mesoscale mechanism to explain the contribution of PNRs. The merit of the proposed mechanism is that it can explain a large number of piezoelectric and dielectric behaviors on mesoscale for relaxor-PT crystals, e.g., high piezoelectric response compared to classical ferroelectric crystals, the temperature-dependent piezoelectric responses, the electric-field-induced polarization/strain, the low polarization hysteresis and dielectric loss at room temperature, etc.

We agree with the referee that our model may not be unique. Indeed, the model we proposed is a mesoscale model which does not provide information about the atomistic mechanisms of PNRs' contribution (e.g., the effects of PNRs on various phonon modes), and hence atomic scale calculations and the phonon experiments at low temperature are still essential for relaxor-PT systems to understand the atomic scale mechanisms. At present, the origin of PNRs is still an open question on the atomic scale, which highly impede the exploration of PNRs' contribution to piezoelectric response on this scale. Therefore, much work is still required for fully clarifying the contributions of PNRs. From this respect, we hope our low-temperature dielectric/piezoelectric experiments and mesoscale simulations could attract considerable attentions from atomic-scale calculation and experimental communities. We demonstrated this viewpoint in the conclusion part of the revised paper:

“It should be noted that on the atomic scale, the origin of PNRs is still an open question, and further in-depth research may be essential for clarifying the contribution of PNRs to dielectric and piezoelectric responses on the atomic scale.”

Comment: (5) Early on they mention a relationship between the PNRs and a softening of the transverse acoustic phonons in Ref. [21]. However, they never explain this known relationship. A softening of the shear mode relates directly to the macroscopic polarization rotation, and hence the ultrahigh piezoelectric response. In Ref. [21] they show that an alignment of the PNRs along [111] softens the [110]-TA phonon. The softening of this phonon at both long and short wavelengths implies a modification of the shear stiffness across multiple length scales. Can the model explain this behavior?

Reply: Thanks for the question. Yes, our model may offer a plausible explanation this behavior.

In Ref. 21 (in the revised paper, it is Ref. 23), the authors showed that the PNRs could significantly affect the structural properties of relaxor-PT crystals. They suggested that the phase instability (softening of TA mode) induced by the PNR–phonon interaction may contribute to the ultrahigh piezoelectric response of relaxor-PT crystals. As the referee mentioned, a softening of the shear mode related directly to the macroscopic polarization rotation. The ease of polarization rotation indicates a high shear piezoelectric response of single domain crystals and a high longitudinal piezoelectric response of domain-engineered crystals.

In our model, we proposed that the PNRs could facilitate the polarization rotation since PNRs are easy to be rotated by a shear stress (Fig. S15 in Supplementary information) and/or a perpendicular electric field (Fig. 5 in main paper). This is equivalent to the softening of TA mode by the existence of PNRs.

Based on the above explanation, we added the following discussion in the revised paper.

“It is worth contrasting our study with previous phonon studies on the contribution of PNRs.²³ Phonon studies showed that the PNR-phonon interaction can induce a phase instability (softening of the TA mode) in relaxor-PTs, thus this phenomenon was thought to play an important role in the ultrahigh piezoelectric response.²³ In our model, it was proposed that PNRs facilitate polarization rotation and enhance the shear piezoelectric response of the single domain state, as shown in Fig. 5 and Fig. S15 of Supplementary information. This suggests a possible connection between the softening of the TA mode (shear) and the PNRs.”

Response to referee #2

Comment: The authors present compelling dielectric and piezoelectric data that suggests that the strong enhancement of the transverse permittivity on heating below ~ 200 K is a prerequisite for the ultrahigh piezoelectricity observed in relaxor ferroelectrics. The authors support their idea with data measured on three different, single-domain, relaxor systems: PMN-0.28PT, PMN-0.32PT, and PZN-0.15PT.

The low-temperature enhancement of the transverse dielectric permittivity is, in each case, accompanied by a significant frequency dispersion, suggesting that the relaxor character is essential. As polar nano-regions (PNR) are widely viewed as underlying relaxor behavior, the conclusion is drawn that the PNR are responsible for the enhancement, and thus the bulk of the ultrahigh piezoelectricity. The authors further support their claim using phase field simulations.

In general, I am favorably inclined to recommend this manuscript for publication. Relaxors, and in particular MPB compositions, are of extremely high interest within the physics and materials science communities. However, before doing so I have several questions:

Reply: Thanks for the positive comments. The low-temperature dielectric relaxation behavior of single domain crystals has never been reported, and the results are original and very important, which will benefit the understanding of the macroscopic properties and the exploration of new relaxor-based material systems. The data provided us a new impetus to analyze the contribution of PNRs. We believe that these data can stimulate considerable attention in the ferroelectric community.

Comment: (1) PZN-0.15PT is a composition that lies on the Ti-rich/tetragonal side of the MPB. Thus I would not have expected it to exhibit relaxor character. The presence of PNR is most directly manifested by the appearance of x-ray or neutron diffuse scattering, which reflects local/short-range structural correlations. As shown by Stock et al., Phys. Rev. B 73 064107 (2006), and Matsuura et al., Phys. Rev. B 74, 144107 (2006), this diffuse scattering vanishes for PMN-xPT compositions on the tetragonal side of the MPB. Can the authors offer a way to reconcile their findings with these two studies?

Reply: Thanks for the comment and these references. We explained the findings reported in the suggested references and compared to our experimental results, as shown in the following.

In the reference [Matsuura et al., Phys. Rev. B 74, 144107 (2006)], the authors studied PMN-0.60PT crystals and found the diffuse scattering was very weak, which implied that the PNRs were not present. We believe that PNRs cannot be detected in PMN-0.60PT, because PT content (60%) is far away from MPB composition (PT content: 30%~35%). In PMN-0.60PT, the long-range ferroelectric behavior should be very strong. At this condition, even there are PNRs induced by the local chemical inhomogeneity, they will be aligned by ferroelectric matrix (in collinear state) due to the interaction between PNRs and ferroelectric matrix. Therefore, the diffuse scattering in PMN-0.60PT is very weak and cannot be observed.

In the reference [Stock et al., Phys. Rev. B 73 064107 (2006)], the authors demonstrated that they observed diffuse scattering for PMN-0.40PT (“We have observed the diffuse scattering just above T_C or T_C^{loc} for PMN-xPT up to $x=40\%$.”). The PZN-0.15PT is similar to PMN-0.40PT, since these

two compositions lie on the tetragonal side of the phase diagram and on the proximity of MPB composition. Fig. R11(a) shows the temperature dependence of the reciprocal of relative dielectric constant for PZN-0.15PT crystal. It can be seen that the dielectric response does not obey Curie-Weiss Law at the temperature above T_C , demonstrating that relaxor character is still present in PZN-0.15PT crystal and PNRs contribute to the dielectric response at the temperatures above T_C . The relaxor behavior was analyzed by a modified Curie law, as shown in Fig. R11(b). In this equation, ϵ_m is the maximum value of the dielectric permittivity at T_m , C is the Curie-like constant, and γ ($=1\sim 2$) is the degree of diffuseness. A higher value of γ represents a higher degree of relaxor behavior. For classical ferroelectrics, the value of γ is 1, while for pure relaxors, the value is 2. The measured γ value of the PZN-0.15PT crystal is 1.5, as shown in Fig. R11(b).

In this reference [Stock et al., Phys. Rev. B 73 064107 (2006)], the authors also observed that diffuse scattering disappeared below T_C for PMN-0.30PT and PMN-0.40PT (“We have observed that the diffuse scattering disappears below T_C for PMN-30%PT and PMN-40%PT.”). They showed the experimental data in the temperature ranges of 340 K $\sim T_C$ and 430 K $\sim T_C$ for PMN-0.30PT and PMN-0.40PT, respectively, revealing that PNRs are not detected over the studied temperature ranges. These results are not in contradiction to our simulation work either. As shown in Fig. 4a of our main paper, all PNRs are aligned by ferroelectric matrix at elevated temperature (350 K $\sim T_C$). At this condition, it is expected that PNRs cannot be detected and distinguished from ferroelectric matrix. Based on our dielectric measurements and phase-field simulation, we think that the diffuse scattering behavior will become evident with further decreasing the temperature, since the PNRs will be gradually isolated from the matrix. Actually, Wen *et al* observed strong diffuse scattering for PMN-0.32PT crystals (similar to PMN-0.30PT) in the temperature range of 300-350 K [Appl Phys Lett 93, 082901 (2008)]. For tetragonal PZN-0.15PT crystals, according to the dielectric measurements and phase-field simulations, we think that the strong diffuse scattering can be observed at temperatures below 200 K. At present, it is a pity that we didn’t find any low-temperature diffuse scattering publication performed on PZN-0.15PT crystals.

In order to show the relaxor behavior of PZN-0.15PT crystals, we added Fig. R11 to Supplementary information (Fig. S3) in the revised paper.

Fig. R11 (a) Temperature dependence of the reciprocal of relative dielectric constant for [001]-oriented PZN-0.15PT crystal. (b) $\log(1/\epsilon_r - 1/\epsilon_m)$ vs. $\log(T - T_m)$ figure at temperatures above T_m for obtaining the parameter γ .

Comment: (2) If "the impact of PNRs on the longitudinal dielectric response is minimal", then what mechanism is responsible for the very large values of d_{33} reported by Guo et al, J. Phys. C 15, L77 (2003) (Fig. 5) for [001] and [110]-oriented single crystals of PMN-xPT near the MPB? Indeed, the data of Guo et al. seem to contradict the authors statement on page 3 that: "... the transverse dielectric and shear piezoelectric responses are significantly larger than their longitudinal counterparts, and thus are the dominant factors for the high performance of relaxor-PT crystals [25-27]." Please comment on this.

Reply: Thanks for the comment and sorry for the confusion. When we claimed that PNRs had a minimal impact on the longitudinal dielectric response, we referred to the longitudinal dielectric response of a single domain state. The "contradiction" mentioned by the reviewer arises from the definitions of piezoelectric coefficients in a single domain and domain-engineered crystals. The crystals poled along nonpolar directions are in a multi-domain state and so-called domain engineered crystals [S. E. Park and T. R. ShROUT, J. Appl. Phys. 82, 1804 (1997); S. J. Zhang and F. Li, J. Appl. Phys. 111, 031301 (2012)]. Generally, the reported high longitudinal d_{33} is for the domain-engineered crystals, which is attributed to the ultrahigh shear piezoelectric $d_{15/24}$ in a single domain state. Strictly speaking, the d_{33} of domain engineered crystals should be called d_{33}^* . We will illustrate this issue in details:

For a single domain crystal, the dielectric permittivity and piezoelectric coefficients are measured in the standard coordinates, where the longitudinal piezoelectric coefficient d_{33} and shear piezoelectric coefficient $d_{15/24}$ are measured by applying an E-field along and perpendicular to the spontaneous polar direction, respectively, as shown in Fig. R12 (a) and (b). In domain engineered crystals, the longitudinal piezoelectric coefficient d_{33}^* is measured along a nonpolar direction, as schematically shown in Fig. R12(c).

From an intrinsic aspect, the longitudinal piezoelectric coefficient d_{33}^* (in an engineered domain state) is determined by the piezoelectric coefficients of a single domain state. Taking a tetragonal crystal as an example, the longitudinal piezoelectric coefficient d_{33} as a function of angle θ away from the polar axis is expressed as [Davis M *et al*, J. Appl. Phys. 101, 054112 (2007)]:

$$d_{33}^* = (\cos \theta \sin^2 \theta)d_{31} + (\cos \theta \sin^2 \theta)d_{15} + d_{33} \cos^3 \theta \quad (1)$$

where d_{ij} are the piezoelectric coefficients of a single domain crystal measured in the standard coordinates. It can be seen from Eq. (1) that the maximum d_{33}^* could be present along a nonpolar direction ($\theta \neq 0$) if d_{15} is large enough compared to d_{33} . An important characteristic of relaxor-PT crystals is that the single domain shear piezoelectric responses are significantly larger than their longitudinal counterparts. As shown in Fig. R13, the maximum d_{33}^* of tetragonal crystals is not present along the polar $\langle 100 \rangle$ directions. Thus, in order to achieve high d_{33}^* , relaxor-PT crystals is generally poled along specific nonpolar directions. For example, tetragonal crystals are poled along [011] or [111] directions, and rhombohedral crystals are poled along [001] or [011] directions. It should be noted that the high d_{33}^* in domain engineered crystals originates from the high shear piezoelectric response in the single domain state.

As discussed above, the piezoelectric coefficient d_{33} in our paper is from single domain crystals, while the reported d_{33} (actually, should be d_{33}^*) by Guo et al is the data from domain-engineered

crystals. Meanwhile, the high d_{33}^* in domain-engineered crystals originates from the high shear piezoelectric coefficient d_{15} of a single domain state.

In the revised paper, we added the following sentence to clarify this issue.

“It should be noted that the large longitudinal piezoelectric properties in domain-engineered relaxor-PT crystals originate from the high shear piezoelectric response in the corresponding single domain state²⁷⁻²⁹.”

Meanwhile, we added a description in the Method section. This description gave the standard coordinate systems for measuring the single domain dielectric and piezoelectric coefficients.

“The single-domain dielectric and piezoelectric coefficients are measured in the standard coordinate systems of single domain crystals. For rhombohedral, orthorhombic and tetragonal single domain crystals, the standard coordinate systems are X: $[1\bar{1}0]$ × Y: $[11\bar{2}]$ × Z: $[111]$, X: $[0\bar{1}1]$ × Y: $[100]$ × Z: $[011]$, and X: $[100]$ × Y: $[010]$ × Z: $[001]$, respectively.”

Fig. R12 Schematics for (a) longitudinal and (b) shear piezoelectric responses of single domain crystals, and (c) longitudinal piezoelectric response of domain engineered crystals. Red arrows represent the spontaneous polarizations of crystals.

Fig. R13 Orientation dependence of piezoelectric coefficient d_{33}^* for a tetragonal PIN-PMN-PT crystal. (a) and (b) are 3D and 2D figures, respectively.

Comment: Other than this, there are minor grammatical mistakes throughout the paper that should be corrected by a native speaker of English.

Reply: Thanks for the suggestion. This paper has been gone through by a native English speaker, we hope the English of the revised manuscript is improved.

Response to referee #3

Comment: In this manuscript the authors attempt to disentangle the question of the “Origin of ultrahigh piezoelectric responses in relaxor-based ferroelectrics” and proceed quite successfully on the basis of a phenomenological description centrally involving the activity of polar nanoregions (PNR) in several relaxor crystals of PMN-PT and PZN-PT.

Reply: We thank the referee for the positive comment.

Here, we would like to clarify that PMN-PT and PZN-PT are relaxor-ferroelectric solid solution crystals, rather than pure relaxor crystals. For classical relaxor PMN, its macroscopic symmetry is cubic and thus has zero piezoelectric response, even with the existence of PNRs. The important microscopic characteristic of relaxor-PT solid solution is the coexistence of PNRs and long-range ferroelectric domains. This distinction between classical relaxor and relaxor-ferroelectric solid solution is important for the referee to re-evaluate our work. The focus of our paper is on the ultrahigh piezoelectricity of relaxor-PT solid solution crystals.

For relaxor-PT solid solutions, current understanding of their high piezoelectricity is focused on the morphotropic phase boundary (MPB). PNR has been thought to play an important role on piezoelectricity of relaxor-PTs, but evidences from piezoelectric/dielectric measurements are still inadequate in substantiating their contribution.

It is the first time, in this paper, we provided important evidence to prove the presence and level of PNR's contribution to piezoelectricity in relaxor-PT crystals by using cryogenic measurements. The low-temperature dielectric relaxation behavior observed in the present work is the most important aspect of this paper and is the key for clarifying the ultrahigh piezoelectricity in relaxor-PT systems. Furthermore, our statement about the contribution of PNRs was strongly supported by the phase-field simulations.

Comment: The authors' final statement “... the contribution of PNRs to the dielectric and piezoelectric response in relaxor-PT crystals can be clearly elucidated,” sounds very optimistic and seems to denote a breakthrough. Unfortunately this promise does not hold at a closer look.

Reply: Thanks for the comment. In this sentence, we were referring to our successful quantification of the PNRs' contribution to the dielectric and piezoelectric responses of relaxor-PT solid solution crystals, and the successful explanations of the contributions by phase-field simulations. Corresponding changes have been made in the manuscript to avoid any misunderstanding or misconception by the readers.

In the revised paper, the sentence “the contribution of PNRs to the dielectric and piezoelectric response in relaxor-PT crystals can be clearly elucidated” was changed to: “Based on the experimental and phase-field simulated results, the contribution of PNRs to the dielectric and piezoelectric properties in relaxor-PT crystals can be explained on the mesoscale.”

Comment: Although the keyword “PNR” has 93 times quoted, the origin of these fancy “polar nanoregions” has not at all been physically explained or theoretically deduced. Instead, the very understanding of the physics behind the PNR has remained highly doubtful.

Reply: We totally agree with the reviewer that the origin of the PNRs in classical relaxor crystals is not well understood despite the vast literature and various models on this topic. However, the focus of our paper is not on the physical origin of PNRs in relaxor crystals. Rather our focus is on the contributions of PNRs to the ultra-high piezoelectricity of relaxor-ferroelectric solid solution crystals and the corresponding mesoscale mechanisms.

Although the origin of PNRs in relaxors is not the focus of this work, we added a short description and corresponding references about the origin of PNRs in the revised paper, as follows:

“Relaxors, e.g., PMN, are characterized by cation order-disorder on the nanoscale¹⁷⁻¹⁸, leading to random fields^{19,20} and local phase fluctuations^{17,18}. These factors lead to a unique characteristic of relaxor-based ferroelectrics in contrast to classical ferroelectrics, i.e., the presence of polar nanoregions (PNRs)¹⁷⁻²¹, which are believed to be responsible for the high dielectric properties of relaxors^{17,18,22}.”

Comment: PNR are simply characterized as “nanoscale inhomogeneities” with “diameters of 3-5 nm, random distribution and volume fraction ~ 7.5%”. In order to stress their apparent segregation from the ferroelectric “host” crystal a novel and uncommon designation of “PNR-ferroelectric composites” has been chosen to replace the long-accepted term “relaxor ferroelectric”.

Reply: We didn’t intend to replace the term “relaxor ferroelectric” using “PNR-ferroelectric composites”. In our work, we modeled the relaxor-PT solid solution (not relaxor PMN itself) as “PNR-ferroelectric composites” based on the fact that the symmetries of the long-range ferroelectric phase (in this case, it is PT) and the local PNRs’ phase (from the presence of PMN or PZN) are different. This description is inspired from the *in situ* microstructure observations of relaxor-PT [Kim KH *et al*, Phys. Rev. B 86, 184113 (2012); Welberry TR *et al*, Phys. Rev. B 74, 224108 (2006); Xu GY *et al*, Phys. Rev. B 70, 174109 (2004); Fu DS *et al*, PRL 103, 207601 (2009)].

Based on XRD and neutron diffuse scattering [Xu GY *et al*, Phys. Rev. B 70, 174109 (2004); Welberry TR *et al*, Phys. Rev. B 74, 224108 (2006); Xu GY, Phys. Rev. B 69, 064112 (2004)], it is known that the symmetry of PNRs is different from that of the ferroelectric matrix, and the size of PNRs is around several nanometers. In our phase field simulations, therefore, we assumed the diameter of PNRs to be 3-5 nm. Meanwhile, the volume fraction of PNRs was set to be in the range of 5% ~ 15%. The calculated temperature dependence of dielectric permittivity for different volume fractions is given in Fig. R14 (Section 3.6 in Supplementary information). The mesoscale mechanism of PNRs’ contribution was found to be similar for different volume fractions. Therefore, we used 7.5% volume fraction as an example. Certainly, the determination of some microscopic parameters of relaxor-PT systems, such as the size and volume fraction of PNRs, is of great interest, but is not related to the main scope of this work.

Fig. R14 Transverse dielectric permittivity ϵ_{11}/ϵ_0 versus temperature for the [100]-poled PNR-ferroelectric composites, with different volume fractions of the PNRs.

Comment: Obviously PNR is considered synonymous to an unknown “defect” and reminds of the very early time of semiconductor physics, where impure materials like Ge and Si showed unexpected phenomena, which were to be understood only many years later. Actually, however, the authors have neglected recent progress of relaxor physics: I. I. K. Jeong et al. have found (Phys. Rev. Lett. 94 (2005) 147602) that “the volume fraction of the PNR [in PMN] as a function of temperature increases from 0% to a maximum of ~30% as the temperature decreases from 650 to 15 K. Below $T \sim 200$ K the volume fraction of the PNRs becomes significant, and PNRs freeze into the spin-glass-like state”. Thus PNR take clearly part in the intrinsic thermodynamics of the relaxor crystal and thus influence many of its properties. In particular, the “spin glass-like state” of the PNR was recently evidenced by W. Kleemann at their percolation limits in SBN and BTZ (Phys. Stat. Sol. (b) 251 (2014) 1993). Relaxors have thus joined the family of “ferroic glasses” like strain und spin cluster glass (X. B. Ren, Phys. Stat. Sol. (b)251 (2014) 1982). According to Jeong et al.(2005)PNR percolation coincides with the superglass transition in PMN at $T_g \sim 239$ K (W.Kleemann, J.Dec, unpublished).

Reply: Thanks for providing the references. As described in our responses above, we didn’t focus on the pure relaxors in this research, thus we didn’t attempt to address the origin of PNRs in the pure relaxors, which has been the focus of many prior efforts as correctly pointed out by the referee. Our main focus of this work is on the contribution of PNRs to the ultrahigh piezoelectric activity of relaxor-ferroelectric solid solution crystals (i.e., PMN-PT and PZN-PT crystals). This is the reason why we did not provide any substantial discussions on the origin of PNRs in relaxors.

According to the reviewer’s suggestion, we cited the recent progress about the origin of PNRs in the revised paper [Kleemann W, Phys. Stat. Sol. (b) 251, 1993 (2014) and references there-in].

Comment: 2. D. Phelan, Z.G. Ye (!), P.M. Gehring et al. (PNAS 111 (2014) 1754) stressed the “Role of random electric fields (RFs) in relaxors” to be “implicated as the genesis of relaxor behavior.” Hence, the authors’ paragraph on p. 10 claiming “the presence of RFs cannot explain the high transverse dielectric and shear piezoelectric response in the relaxor-PT crystals” must be cast severely in doubt. First of all, the RF-assisted creation of static PNR below the Burns temperature $T_d \sim 600$ K has escaped the authors’ modeling. Instead, PNR were taken as defects like dopants in a

semiconductor. Thus they missed the outcome of the intrinsic disorder of heterovalent cations, which creates a “frozen” charge disorder and, hence, random electric fields with many unusual consequences, viz. the essence of the “enigmatic relaxor physics”.

Reply: Thanks for the valuable comment. Following the referee’s comment on “the presence of RFs cannot explain the high transverse dielectric and shear piezoelectric response in the relaxor-PT crystals” could lead to some misunderstandings, we carried out additional phase-field simulations for a tetragonal ferroelectric crystal with consideration of RFs, we found that RFs indeed cannot account for the experimental observations for relaxor-PT crystals, i.e., high transverse dielectric permittivity but very low longitudinal values of the single domain crystals. The detailed simulation results and discussions are given below:

In the phase-field simulation, RFs were added into every grid of a tetragonal crystal, in order to investigate the impact of random field on dielectric response. The RFs were set to obey *Gaussian* distribution. The standard deviation σ was selected to be 0 (no random field), 3 MV/m, 5 MV/m and 10 MV/m in the simulation. The dielectric permittivities vs. temperature at different magnitudes of RFs are given in Fig. R15. It can be observed that the dielectric responses are not significantly enhanced by random fields in the case of $\sigma=3\text{MV}$ and 5MV . For $\sigma=10\text{ MV/m}$, however, both dielectric permittivities ϵ_{11}/ϵ_0 (transverse) and ϵ_{33}/ϵ_0 (longitudinal) were found to be higher than those of $\sigma=0$ for temperatures above 300 K, this is due to the fact that the single domain state has been destroyed and destabilized at this condition, as shown in Fig. R16. Of particular interest is that the impact of RFs is much higher on the longitudinal dielectric permittivity (ϵ_{33}/ϵ_0) than that on the transverse counterpart (shown in Fig. R15), which is not the scenario of our experimental observation.

According to this simulation, one can find that: (1) RFs cannot directly benefit the transverse dielectric and shear piezoelectric properties for a single domain crystal; (2) as the single domain state is destroyed, the contribution of RFs to the longitudinal dielectric permittivity is much larger than that to the transverse one. Therefore, adding mesoscale RFs in a ferroelectric crystal cannot explain the high transverse dielectric and shear piezoelectric property of single domain relaxor-PT crystals.

Fig. R15 The simulated dielectric permittivity vs. temperature for a [100]-poled tetragonal crystal at different magnitudes of RFs. (a) Transverse dielectric permittivity ϵ_{11}/ϵ_0 ; (b) longitudinal dielectric permittivity ϵ_{33}/ϵ_0 .

Fig. R16 Simulated domain structure for a [100]-poled tetragonal crystal with random fields of $\sigma=10$ MV at (a) 300 K and (b) 350 K. The x- and y-axes represent [100] and [010] directions, respectively. The unit for x- and y-axes is nm. The color bar denotes the angle (unit: degree) between the polar vector and the [100] direction.

However, even so, we should admit that phase-field modeling is a mesoscale calculation method, the effect of RFs at the atomic-scale has yet to be considered. Therefore, we cannot exclude the contribution of RFs to piezoelectricity from atomic-scale mechanism(s), for example, RFs assist the formation of PNRs. RFs could be responsible for the high piezoelectricity of relaxor-PT crystals, if RFs play a dominant role in the formation of PNRs in relaxor-ferroelectrics. As we mentioned previously, the aim of this paper is not to discuss the origin of PNRs, and the discussions on the effects of RFs on piezoelectric properties of relaxor-PT crystals are out of the scope of this paper. In the revised manuscript, we deleted the corresponding discussions on RFs and added the following sentence:

“It should be noted that on the atomic scale, the origin of PNRs is still an open question, and further in-depth research may be essential for clarifying the contribution of PNRs to dielectric and piezoelectric responses on the atomic scale.”

Comment: 3. B.-X. Xu, S. Wang, M. Yi (Proc. Appl. Math. Mech. 15 (2015) 723/ DOI 10.1002/pamm.201510348) have considered “A finite element phase field model for relaxor ferroelectrics”. The model is derived from thermodynamic analysis including the material force theory. Random field theory is adopted to take the disorder of relaxor ferroelectrics into account. Results show that the model is capable of reproducing relaxor features, such as domain miniaturization, small remnant polarization and large piezoelectric response. Dependence of these features on the random field strength is in line with experimental experience. Since the present authors’ incomplete attempts of a relaxor phase field theory cannot compete with the last-cited professional one, I see no chance of publication.

Reply: Thanks to the reviewer for offering the reference. However, this reference is on phase-field modeling of relaxor crystals. Pure relaxor PMN does not possess a long-range ferroelectric order and does not possess a remnant polarization, and hence it is non-piezoelectric. This type of phase-field simulations was firstly performed by Semenovskaya, S. & Khachatryan, A. G [Semenovskaya, S. & Khachatryan, A. G., J. Appl. Phys. 83, 5125-5136 (1998); Semenovskaya, S. & Khachatryan, *Ferroelectrics* **206**, 157-180 (1998)].

In our paper, we focused on the relaxor-PT solid solutions with compositions near MPB that have a very strong long-range ferroelectric property and a large macroscopic polarization. As shown in Fig. R17, the systems studied by us and B.X. Xu *et al* are very different. The aim of B.X. Xu *et al* is to reproduce the microstructure and PE/SE loops for the pure relaxors such as PMN and SBN80. The aim of our work is to simulate the abnormal dielectric and piezoelectric behaviors at low-temperature and explore the contribution of PNRs to piezoelectric activity for relaxor-PT solid solution crystals, in which there is strong ferroelectricity but embedded with PNRs.

As for the results of this reference, we believe that a small remnant polarization can be reproduced by adding RFs in a typical ferroelectric crystal, but we are surprised by the “large piezoelectric response” being reproduced. PMN is an electrostrictive material without piezoelectricity, since its macroscopic (average) phase is centrosymmetric.

To the best of our knowledge, there are no existing publications that successfully explained the ultrahigh piezoelectric responses and the abnormal temperature dependent behavior in relaxor-PT systems. In our paper, we firstly provided important evidence to elucidate the contribution of PNRs to piezoelectric activity in relaxor-PT crystals by using cryogenic measurements. And then, we gave a mesoscale mechanism to further explain the contribution of PNRs by phase field simulation. In addition, the phase-field modeling results are also consistent with the microstructural observations of relaxor-PT materials.

In the revised paper, we added this reference and Semenovskaya *et al*'s papers, as shown in below:

“Phase-field method has been employed to model the effects of random defects/fields on ferroelectric domains and domain-switching to simulate the relaxor behavior³⁷⁻⁴⁰.”

Fig. R17 Polarization vs electric field of (a) PMN (classical relaxor) and (b) PMN-0.28PT (relaxor-ferroelectric solid solution with composition near MPB) crystals at room temperature.

Comment: In view of these large deficiencies the manuscript is not acceptable for NComms.

Reply: We believe we have improved our manuscript by taking into account the comments from all three reviewers, and our responses have clarified the misunderstandings. We hope the reviewer is convinced to provide a positive recommendation on the revised manuscript.

Best wishes,

Fei Li, Shujun Zhang, Thomas ShROUT & Long-Qing Chen

Reviewers' Comments:

Reviewer #1 (Remarks to the Author):

I appreciate the work done to address all of my concerns. For all except for one point I am satisfied. The one I am still having problems with is my most significant concern. The fact that the co-linear condition, which is used to explain the ultrahigh piezoelectric properties, is not reflecting the details of PNR diffuse scattering experiments.

In the case of PMN-0.3PT, poling along [100] into the 4R domain state does not change the appearance of the PNR diffuse scattering butterfly pattern around the Bragg peaks at all. Instead it changes a second component of the diffuse scattering. This component is qualitatively different and likely comes from the off centering of Pb atoms. The off centering of Pb atoms along [100] directions was deduced from pair distribution function measurements (T. Egami, S. J. L. Billinge, *Underneath the Bragg Peaks: Structural Analysis of Complex Materials*, R. W. Cahn, Ed. (Pergamon Materials Series, Oxford, 2003)) and predicted from theory (B. P. Burton and E. Cockayne, *Phys. Rev. B* 60, R12542 (1999)). The effect of poling on this second component was also more recently found in diffuse elastic scattering and local dynamics (Manley et al. *Sci. Adv.* 2, e1501814 (2016)). In the domain engineered poled state the piezoelectric response increases, and the material exhibits a shear softening (R. Zhang, B. Jiang, W. W. Cao *J. Appl. Phys.* 90, 3471 (2001)), but the only change in the diffuse scattering structure is in the second component along the [100] poling direction. The problem is that the polar nanoregions themselves are actually multicomponent while the model in this paper only considers a single component.

As far as I can tell, this model is able to explain the case of crystals poled along [111], since the macrodomain and PNR diffuse scattering co-align in the field direction. But it cannot explain the important role of the second component to the polar nanoregions that align with the [100] poling field in the domain engineered crystal. This is the arrangement that is actually used in applications. The fact that the model is unable to capture this important case is a weak point.

Because the second component in the diffuse scattering likely comes from an atomic scale off centering of Pb atoms, it is beyond the mesoscale model used in the paper.

At this point I tend to agree with reviewer 3 that the model is too simple, although for a different reason.

Reviewer #2 (Remarks to the Author):

I am satisfied by the authors revised manuscript and their replies to my comments. Depending on the opinions of the other referees, I can now recommend publication.

I note that the neutron-spin echo study of PMN by Stock et al. (*PRB*, 2010) found a relaxational mode associated with the diffuse scattering (PNR). When these data were fit to an Arrhenius law, the relaxation decay time was well-described using an activation energy $U = 1100 \text{ K} \pm 300 \text{ K}$. This result seems relevant to Comment #4 of the first referee.

Reviewer #3 (Remarks to the Author):

The manuscript has substantially been improved, but still some points remain to be remedied:

line 49: The term "cation order-disorder" does not make sense and must be replaced by "cation disorder".

line 49: The citation "17, 18" must be extended to "17 - 20".

line 61: Replace the word "critical" by "crucial".

line 256: In the fairly general outlook onto other diluted systems at least the remark "and dilute Ising antiferromagnet $\text{Fe}_{0.55}\text{Mg}_{0.45}\text{Cl}_2$ " must be omitted, since the co-existence of antiferromagnetic and spin glass phases in this system is subject to simultaneous percolation of the AF phase with non-percolating superantiferromagnetic clusters forming a superspin glass [see W. Kleemann et al., Phys. Rev. Lett. 105 (2010) 257202 (last sentence), and S. Chillal et al., Phys. Rev. B 87 (2013) 220403(R)]. This has nothing in common with the relaxor problem, which involves not only structural, but in particular charge disorder. Note that this would change in case of an applied magnetic field, where the dilute antiferromagnet also experiences staggered magnetic random fields (J. Cardy, Phys. Rev. B 29 (1984) 505] being equivalent to "quenched staggered magnetic moments".

lines 258 – 260: I strongly recommend omission of the new unsolicited sentences "It should be noted that on the atomic scale, the origin of PNRs is still an open question, and further in-depth research may be essential for clarifying the contribution of PNRs to dielectric and piezoelectric responses on the atomic scale.". These concluding remarks are mourning the authors' lacking knowledge of the sources of PNR at the atomic scale, but simultaneously would falsify the truth, viz. their knowledge of quenched random electric fields, which were explicitly mentioned in the preceding text (lines 49 f.), albeit, without any comment on their own opinion.

After satisfactory comply with these recommendations, I shall be ready to support the publication of this ms.

We thank all three referees for their diligent efforts in carefully reading the manuscripts, making constructive comments, and providing useful suggestions for improving the manuscript. Their comments and suggestions have indeed helped us significantly improve the manuscript. We are happy to learn that both referees #2 and #3 are now recommending publication of the revised manuscript subject to some very minor revisions, while referee #1 is satisfied with all of our responses and revisions except one remaining point. We have revised our manuscript according to the new comments by all the referees, and herewith we address the remaining point raised by referee #1 with regard to reconciling the proposed model with the diffuse scattering experimental observations.

Response to referee #1

Comment: I appreciate the work done to address all of my concerns. For all except for one point I am satisfied.

Reply: We appreciate the positive comment by the referee that we have addressed all his/her comments except one.

The one I am still having problems with is my most significant concern. The fact that the co-linear condition, which is used to explain the ultrahigh piezoelectric properties, is not reflecting the details of PNR diffuse scattering experiments. In the case of PMN-0.3PT, poling along [100] into the 4R domain state does not change the appearance of the PNR diffuse scattering butterfly pattern around the Bragg peaks at all. Instead it changes a second component of the diffuse scattering. This component is qualitatively different and likely comes from the off centering of Pb atoms. The off centering of Pb atoms along [100] directions was deduced from pair distribution function measurements (T. Egami, S. J. L. Billinge, *Underneath the Bragg Peaks: Structural Analysis of Complex Materials*, R. W. Cahn, Ed. (Pergamon Materials Series, Oxford, 2003)) and predicted from theory (B. P. Burton and E. Cockayne, *Phys. Rev. B* 60, R12542 (1999)). The effect of poling on this second component was also more recently found in diffuse elastic scattering and local dynamics (Manley et al. *Sci. Adv.* 2, e1501814 (2016)). In the domain engineered poled state the piezoelectric response increases, and the material exhibits a shear softening (R. Zhang, B. Jiang, W. W. Cao *J. Appl. Phys.* 90, 3471 (2001)), but the only change in the diffuse scattering structure is in the second component along the [100] poling direction. The problem is that the polar nanoregions themselves are actually multicomponent while the model in this paper only considers a single component.

As far as I can tell, this model is able to explain the case of crystals poled along [111], since the macrodomain and PNR diffuse scattering co-align in the field direction. But it cannot explain the important role of the second component to the polar nanoregions that align with the [100] poling field in the domain engineered crystal. This is the arrangement that is actually used in applications. The fact that the model is unable to capture this important case is a weak point.

Because the second component in the diffuse scattering likely comes from an atomic scale off centering of Pb atoms, it is beyond the mesoscale model used in the paper.

At this point I tend to agree with reviewer 3 that the model is too simple, although for a different reason.

Reply: We appreciate the constructive and objective comments and discussions on our proposed model by the referee, we also thank the referee for bringing to our attention a very recent reference: Manley *et al.*, *Sci. Adv.* 2, e1501814 (2016).

We agree with the referee that the polar nanoregions themselves could be multicomponent as observed in diffuse scattering experiments, i.e., a butterfly-shaped pattern around Bragg peaks and a broader component [Manley *et al.*, *Sci. Adv.* 2, e1501814 (2016)]. Our phase-field model is actually

based on the first component, i.e., a butterfly-shaped pattern, since it is thought to be the most important distinction between relaxor-PT crystals and normal ferroelectric crystals [Xu GY *et al*, Phys. Rev. B, **70**, 174109 (2004); Welberry TR *et al*, Phys. Rev. B, **74**, 224108 (2006); Phelan D *et al*, Natl. Acad. Sci. **111**, 1754-1759 (2014).].

We admit that our mesoscale model doesn't specifically address atomic processes such as the off-centering of Pb. However, we believe our proposed mesoscale model in this manuscript is a major step forward in understanding the mesoscale mechanisms for the ultra-high piezoelectric responses of relaxor-ferroelectric solid solution systems. For example, it successfully explains the level of PNR's contribution to dielectric and piezoelectric properties, the corresponding temperature-dependent dielectric and piezoelectric behaviors, and the low piezoelectric hysteresis at room temperature regardless of the ultrahigh piezoelectric properties. Of particular importance is that there is a consensus from the past 20 years of extensive researches that the high longitudinal piezoelectric response of domain-engineered crystals originates from the high shear piezoelectric response in single domain state [Damjanovic D, *IEEE Trans Ultrason Ferroelectr Freq Contr*, **56**, 1574-1585 (2009); Zhang, S. *et al*, *J Appl Phys*, **111**, 031301 (2012); Sun E. W. *et al*, *Prog Mater Sci*, **65**, 124-210 (2014), and references therein]. Therefore, we believe our assumption that the mechanism of the high shear piezoelectricity in single domain crystals is the same as the high longitudinal piezoelectricity in the engineered-domain cases is justified. We also proved this concept by phase-field simulation and temperature-dependent dielectric measurements in domain-engineered PZN-0.15PT crystals, as discussed in previous response letter.

In fact, the main conclusions from our work, i.e. the high shear piezoelectricity and the polarization rotation in relaxor-PT crystals are generated and facilitated by the presence of PNRs, do not contradict the main arguments made in the new reference based on diffuse scattering experiments mentioned by the referee [Manley *et al*. Sci. Adv. 2, e1501814 (2016)]: "Our results reveal that domain engineering of relaxor-based ferroelectric single crystals (31) enhances the electromechanical coupling for two reasons, one deliberate and one serendipitous. Whereas the deliberate 4R arrangement of the domains mechanically facilitates the conversion of the high-shear piezoelectric response into a longitudinal response (31), the serendipitous alignment of the PNR vibrational modes and local structure (Fig. 3) further enhances this shear response by softening the TA phonon through the anticrossing (Fig. 1F)." The difference between our work and this reference only lies on how PNRs impact the shear response.

As to the second component of diffuse scattering that is observed when a PMN-30PT crystal is poled along [100] into the 4R domain state, we agree with the referee that it likely comes from the off-centering of Pb cations along the [100] directions. This can be explained as follows. In PMN-30PT, the nature of PNRs is mainly controlled by their Landau potential and local fields (i.e., local electric field, elastic field, and gradient driving field due to the discontinuity of polarization and strain around the interfaces between PNRs and ferroelectric matrix). It is observed in phase-field simulations that the local electric fields ($10\sim 10^2$ MV/m) are generally one or two orders larger than a practical poling electric field (1 MV/m), since the size of PNRs is only several nanometers. Therefore, poling along [100] would not change the appearance of the PNR diffuse scattering butterfly pattern around the Bragg peaks, but instead it may trigger polarization rotation of some macro- or micro-ferroelectric domains (facilitated by the initial presence of PNRs, as documented in this work) which necessarily involves the off-center displacements of Pb cations, since Pb^{2+} with its lone electron pair is undoubtedly the major contributor to the polarization.

The proposed role of off-centering of Pb atoms in the piezoelectric responses cited in the reference by the referee is interesting, and we believe this hypothesis can be tested at low-temperatures. As shown in Fig. R1, a drastic decrease of the longitudinal dielectric and piezoelectric properties was observed for the [001]-poled PMN-0.30PT crystals at temperatures below 100 K, thus it is expected

that a significant change of the second component could be observed in this temperature range if it is associated with the high piezoelectricity. This test is beyond the scope of this work, but it would be rewarding to do it in the near future.

Therefore, while the referee may not fully agree with our arguments and may still not be 100% satisfied with our responses, we hope the referee will agree with us that publication of this work will attract significant new efforts in this area to fully demystify the role of PNRs in the ultra-high piezoelectric responses of relaxor-ferroelectric solid solutions at both the mesoscale and atomic scale. We also hope the referee will agree with us that for the benefits of future efforts, it is essential to publish the important results from different parties although the results might not agree with each other 100% at this stage. We strongly believe that future research in this community will lead to fully consistent models and mechanisms at both atomic and mesoscales.

Taking into account the referee's comments and suggestions, in the revised paper, we have cited the new reference [Manley et al. *Sci. Adv.* 2, e1501814 (2016)]. We believe it will help the readers of our paper understand there are other possible mechanisms for the ultrahigh piezoelectricity of relaxor-PT crystals. We have also added the following sentence in the *Introduction* section of the revised paper: "For example, Xu *et al.*²³ proposed that the softening of the transverse acoustic mode was due to the existence of PNRs, while M. E. Manley *et al.*²⁴ further demonstrated that aligning PNR vibrational modes by a poling electric field can enhance the phonon softening."

Fig. R1 Temperature-dependent longitudinal dielectric permittivity ϵ_{33}^* (a) and piezoelectric coefficient d_{33}^* (b) of a [001]-poled rhombohedral PMN-0.30PT crystal. (Unpublished data)

Response to referee #2

Comment: I am satisfied by the authors revised manuscript and their replies to my comments. Depending on the opinions of the other referees, I can now recommend publication.

I note that the neutron-spin echo study of PMN by Stock et al. (*PRB*, 2010) found a relaxational mode associated with the diffuse scattering (PNR). When these data were fit to an Arrhenius law, the relaxation decay time was well-described using an activation energy $U = 1100 \text{ K} \pm 300 \text{ K}$. This result seems relevant to Comment #4 of the first referee.

Reply: We thank the referee for his/her positive recommendation. We also thank the referee for providing the reference by Stock *et al*, which is very useful to our future investigations.

Response to referee #3

Comment: The manuscript has substantially been improved, but still some points remain to be remedied:

Reply: We appreciate the positive comment by the referee. We have further revised this paper according to the referee's suggestions as described in the responses below.

Comment 1: line 49: The term "cation order-disorder" does not make sense and must be replaced by "cation disorder".

Reply: Thanks for the suggestion. We have replaced "cation order-disorder" by "cation disorder" in the revised manuscript.

Comment 2: line 49: The citation "17, 18" must be extended to "17 - 20".

Reply: Thanks for the suggestion. The citation has been extended to "17 - 20".

Comment 3: line 61: Replace the word "critical" by "crucial".

Reply: Thanks for the suggestion. The word "critical" has been replaced by "crucial" in the revised paper.

Comment 4: line 256: In the fairly general outlook onto other diluted systems at least the remark "and dilute Ising antiferromagnet $\text{Fe}_{0.55}\text{Mg}_{0.45}\text{Cl}_2$ " must be omitted, since the co-existence of antiferromagnetic and spin glass phases in this system is subject to simultaneous percolation of the AF phase with non-percolating superantiferromagnetic clusters forming a superspin glass [see W. Kleemann et al., Phys. Rev. Lett. 105 (2010) 257202 (last sentence), and S. Chillal et al., Phys. Rev. B 87 (2013) 220403(R)]. This has nothing in common with the relaxor problem, which involves not only structural, but in particular charge disorder. Note that this would change in case of an applied magnetic field, where the dilute antiferromagnet also experiences staggered magnetic random fields (J. Cardy, Phys. Rev. B 29 (1984) 505] being equivalent to "quenched staggered magnetic moments".

Reply: Thanks for the suggestion and the references. In the revised paper, we have deleted the remark about "dilute Ising antiferromagnet $\text{Fe}_{0.55}\text{Mg}_{0.45}\text{Cl}_2$ ".

Comment 5: lines 258 – 260: I strongly recommend omission of the new unsolicited sentences "It should be noted that on the atomic scale, the origin of PNRs is still an open question, and further in-depth research may be essential for clarifying the contribution of PNRs to dielectric and piezoelectric responses on the atomic scale.". These concluding remarks are mourning the authors' lacking knowledge of the sources of PNR at the atomic scale, but simultaneously would falsify the truth, viz. their knowledge of quenched random electric fields, which were explicitly mentioned in the preceding text (lines 49 f.), albeit, without any comment on their own opinion.

Reply: Thanks for the suggestion. In the revised paper, the corresponding sentence has been deleted.

Comment 6: After satisfactory comply with these recommendations, I shall be ready to support the publication of this ms.

Reply: We thank the reviewer for his/her positive recommendation.

Fei Li, Shujun Zhang, Zuo-Guang Ye, Thomas R. Shroud and Long-Qing Chen

Reviewers' Comments:

Reviewer #1 (Remarks to the Author):

The authors made substantial improvements to the manuscript. I am now satisfied and recommend publication in Nature Communications.

Reviewer #3 (Remarks to the Author):

The authors took account of all of my previous suggestions. As promised, I am now in favor of getting this manuscript published. It might be considered as a useful platform for future discussions about the very nature of relaxor ferroelectrics and their still enigmatic PNRs